



# Assessing and zoning of typhoon storm surge risk with GIS technique: A case study of the coastal area of Huizhou

Si Wang[1], Lin Mu[1,2], Zhenfeng Yao[3], Jia Gao[1], Enjin Zhao[1]

[1]College of Marine Science and Technology, China University of Geosciences, Wuhan, 430074, China
5  [2]Shenzhen Research Institute, China University of Geosciences, Shenzhen, 518057, China
[3]Department of Natural Resources of Huizhou Bureau, Huizhou, 516001, China

*Correspondence to*: Lin Mu (e-mail: moulin1977@hotmail.com).

**Abstract.** Storm surge is one of the most destructive marine disasters to life and property for Chinese coastal regions, especially for Guangdong province. In Huizhou city, Guangdong province, due to the high concentration of chemical and 10  petroleum industries and the high population density, the low-lying coastal area is susceptible to the storm surge. Therefore, a comprehensive risk assessment of storm surge over the coastal area of Huizhou can delimit zones that could be affected to reduce disaster losses. In this paper, typhoon intensity for the minimum central pressure of 880 hPa, 910 hPa, 920 hPa, 930 hPa, and 940 hPa (corresponding to 1000-year, 100-year, 50-year, 20-year, and 10-year return period) scenarios were designed to cover possible situations. The Jelesnianski method and the Advanced Circulation (ADCIRC) model coupled with 15  the Simulating Waves Nearshore (SWAN) model were utilized to simulate inundation extents and depths of storm surge over the computational domain under these representative scenarios. Subsequently, the output data from the coupled simulation model (ADCIRC–SWAN) were imported to Geographical Information System (GIS) software to conduct the hazard assessment for each of the designed scenarios. Then, the vulnerability assessment was made based on the dataset of land cover types in the coastal region. Consequently, the potential storm surge risk maps for the designed scenarios were 20  produced by combining hazard assessment and vulnerability assessment with the risk matrix approach. The risk maps indicate that due to the protection given by storm surge barriers, only a small proportion of the petrochemical industrial zone and the densely populated communities in the coastal areas were at risk of storm surge for the scenarios of 10-year and 20-year return period typhoon intensity. Moreover, some parts of the exposed zone and densely populated communities were subject to high and very high risk when typhoon intensities were set to a 50-year or a 100-year return period. Besides, the 25  scenario with the most intense typhoon (1000-year return period) induced the very high risk to the coastal area of Huizhou. Accordingly, the risk maps can help decision-makers to develop risk response plans and evacuation strategies in coastal communities with the high population density to minimize civilian casualties. The risk analysis can also be utilized to identify the risk zones with the high concentration of chemical and petroleum industries to reduce economic losses and prevent environmental damage caused by the chemical pollutants and oil spills from petroleum facilities and infrastructures 30  that could be affected by storm surge.



## 1 Introduction

Storm surge is the abnormal rise in sea level during tropical cyclones, and the surge is primarily produced by strong storm winds pushing water into shore. When tropical cyclone makes landfall, the accompanying storm surge will lead to significant

flooding in the surrounding coastal area. Therefore, storm surge associated with tropical cyclones is a devastating hazard and frequently causes considerable deaths and property damage in many coastal regions of the world. For example, in August 2005, Hurricane Katrina struck the United States, and the storm surge from Katrina along the Gulf coastal area was ranging from 10 to 28 feet high. It caused 108 billion dollars in property damages and resulted in a death toll of 1833 persons (National Oceanic and Atmospheric Administration, 2016). In 2019, typhoon Hagibis caused server storm surge flooding

that inundated southern areas of Honshu and killed at least 99 people in Japan (Asian Disaster Reduction Center, 2019).

In China, storm surge is regarded as one of the most seriously marine disasters, which inflicts tremendous losses to people's life and property. Every year, about 27 tropical cyclones are formed over the western North Pacific Ocean and the South China Sea, and one-fourth of them strike directly coastal regions of China. The tropical cyclone storm surge, from 2000 to 2018, caused a direct economic loss of 10.82 billion yuan and left a death toll of 46 on average annually in China

(Ministry of Natural Resources of the People's Republic of China, 2018). These economic damages and casualties will be further increased because of the explosive population growth and the rapid development of society in the coastal area of China (Seto et al., 2013; Lichter et al., 2011; McGranahan et al., 2007). Moreover, with considerably rising sea levels caused by continued global warming, tropical cyclone-induced storm surge will be more destructive in the future (Intergovernmental Panel on Climate Change, 2018). Therefore, it is important to establish storm surge preparedness plans in advance to reduce

economic losses and casualties for coastal cities.

Huizhou is the eastern city in the Pearl River Delta region in Guangdong province, China. In Huizhou, the petrochemicals and electronic information have developed into the dominant industries, with an annual production value of more than one trillion yuan. The Daya Bay Petrochemical Zone, located in the coastal area of Huizhou, is currently ranked the first in China in terms of the scale of integrating refinery and petrochemical production. The high concentration of petroleum refining

facilities and energy infrastructures in the Zone and the high density of population in low-lying coastal communities make the coastal area of Huizhou especially vulnerable to storm surges. Moreover, storm surge risk that coastal area of Huizhou faces could be increased with sea-level rise, population growth, and further petrochemical industry development. Therefore, it is necessary to assess the potential risk of typhoon storm surge in advance in Huizhou to help decision-makers understand the affected regions and allow them to develop mitigation strategies and future land use planning.

The terminologies and methods for coastal risk assessment vary in different scholars and organizations. In most case, the risk assessment of storm surge flooding is determined by the combination of hazard, exposure, and vulnerability (Chrichton, 1999; Kaźmierczak and Cavan, 2011; Koks et al., 2015):

(1). The hazard is defined as a natural event that causes impacts to people and infrastructures. The hydrodynamic models and wave models with statistical methods are applied to the qualitative evaluation of storm surge hazard. With the statistical





methods, the relation between minimum central pressure and return period of typhoons can be constructed through historical storm records at specific zones. Then, the hydrodynamic models and the wave models are utilized to simulate potential inundation extents and depths under different typhoon intensity scenarios.

(2). The exposure means the elements at risk to a given hazard event. The exposure assessment of storm surge is performed by measuring the counts of affected elements in the inundated region.

(3). The vulnerability refers to the degree to which a natural element is susceptible to sustaining damage from a hazard event. The empirical stage-damage curve is a common method for vulnerability assessments. In order to develop stage-damage curves for elements, researchers need to observe and analyze the damage of elements in the disaster-affected area and assign the percentage damage to different types of elements according to the scale of damage and disaster degree.

The quantitative stage-damage curve method for vulnerability assessment (Middelmann, 2010; de Moel et al., 2014;
McGrath, 2015) is common in developed countries. However, because of lacking adequate damage-cost data collected through post-disaster surveys, constructing a stage-damage curve is difficult in developing countries. Thus, the stage-damage curve method for vulnerability assessment is not appropriate for Chinese coastal regions. This limited data availability has led to other data utilized to assess vulnerability in China. In 2019, a panel of Chinese ocean disaster prevention and reduction researchers published the latest *standard Technical guideline for risk assessment and zoning of marine disaster (Part 1:*
*storm surge)* (Ministry of Natural Resources of the People's Republic of China, 2019). The standard *guideline* provides the procedures and approaches for hazard, vulnerability, and risk assessment of storm surge (Section 3). According to the standard *gu*ideline, the land cover types and corresponding vulnerability values rather than the stage-damage curve method are recommended to conduct a vulnerability assessment in China.

In the past few years, the standard *guideline* has been taken and put into operation of storm surge risk assessment in the
coastal areas of China (Fang et al., 2016; Zhang et al., 2016). However, few studies have made a risk assessment of storm surge in Huizhou city. This paper attempts to conduct a comprehensive assessment and zonation of storm surge risk in the coastal area of Huizhou city with the latest standard *guideline*. First, the relation between minimum central pressure and return period was constructed from historical tropical cyclone records with statistical approaches in Huizhou. Then, the typhoon intensities for the minimum central pressure of 880 hPa, 910 hPa, 920 hPa, 930 hPa, and 940 hPa (corresponding to
1000-year, 100-year, 50-year, 20-year, and 10-year return period) scenarios were designed. Subsequently, the Jelesnianski method was utilized to generate wind field and the Advanced Circulation (ADCIRC) hydrodynamic model coupled with the Simulating Waves Nearshore (SWAN) wave model were employed to simulate the storm surge for each of the designed typhoon intensity scenarios. The data of simulated storm surge over the computational domain under different typhoon intensity scenarios were generated from the coupled ADCIRC+ SWAN model. Afterward, the inundation extents and depths
of storm surge maps can be created from the data in Geography Information System (GIS) integrated software to visualize and assess storm surge hazard levels. Eventually, combining hazard assessment with vulnerability assessment, the risk maps during a specific typhoon intensity in the study area were obtained. The risk maps can help decision-makers to develop evacuation plans for the densely populated communities in the coastal area. The risk analysis can be utilized to identify the





risk zones with the high concentration of chemical and petroleum industries to prevent environmental damage caused by the
chemical pollutants and oil spills from affected petroleum industries.

The rest of the paper is organized as follow: the study area and datasets are described in section 2; section 3 details the methodology and procedure including the model description and validation, storm parameters and scenarios design, and the procedure for risk assessment; results and discussion are stated in section 4; the conclusion of the study is drawn and future research is provided in section 5.

**2 Study area and datasets**

**2.1 Study area**

Guangdong is a coastal province, located in the southernmost part of China, and has a long coastline along the South China Sea, as shown in Figure 1 (a) and Figure 1 (b). Guangdong is one of the most prosperous provinces in China with the highest GDP of 9.73 trillion yuan and a population of 113.46 million in 2018. However, due to the geographical position,
Guangdong is the most frequently affected province by tropical cyclones in China. The storm surge is regarded as the most seriously marine disaster for the Guangdong province. During the period from 1949 to 2017, 263 tropical cyclones landed in Guangdong province (Ying et al., 2014). In 2018, 3 different typhoons (Ewiniar, Bebinca, and Mangkhut) made landfall on Guangdong province, which left 4 people dead and caused a direct economic loss of 2.37 billion yuan.

The Huizhou city is located in the southeastern area of Guangdong province, and it occupies part of the Pearl River Delta
megalopolis to the northeast of Hong Kong and Shenzhen. It, spanning from 22˚4'N to 23˚57'N latitude and 113˚51'W to 115˚28'W longitude, covers a land area of about 11347 km$^2$ and sea area of approximately 4520 km$^2$. There are two districts (Huicheng and Huiyang) and three counties (Boluo, Huidong, and Longmen) in Huizhou, as shown in Figure 1 (c). The coastal region of Huiyang district and Huidong county have been affected by the tropical cyclones during the season running from April to November.


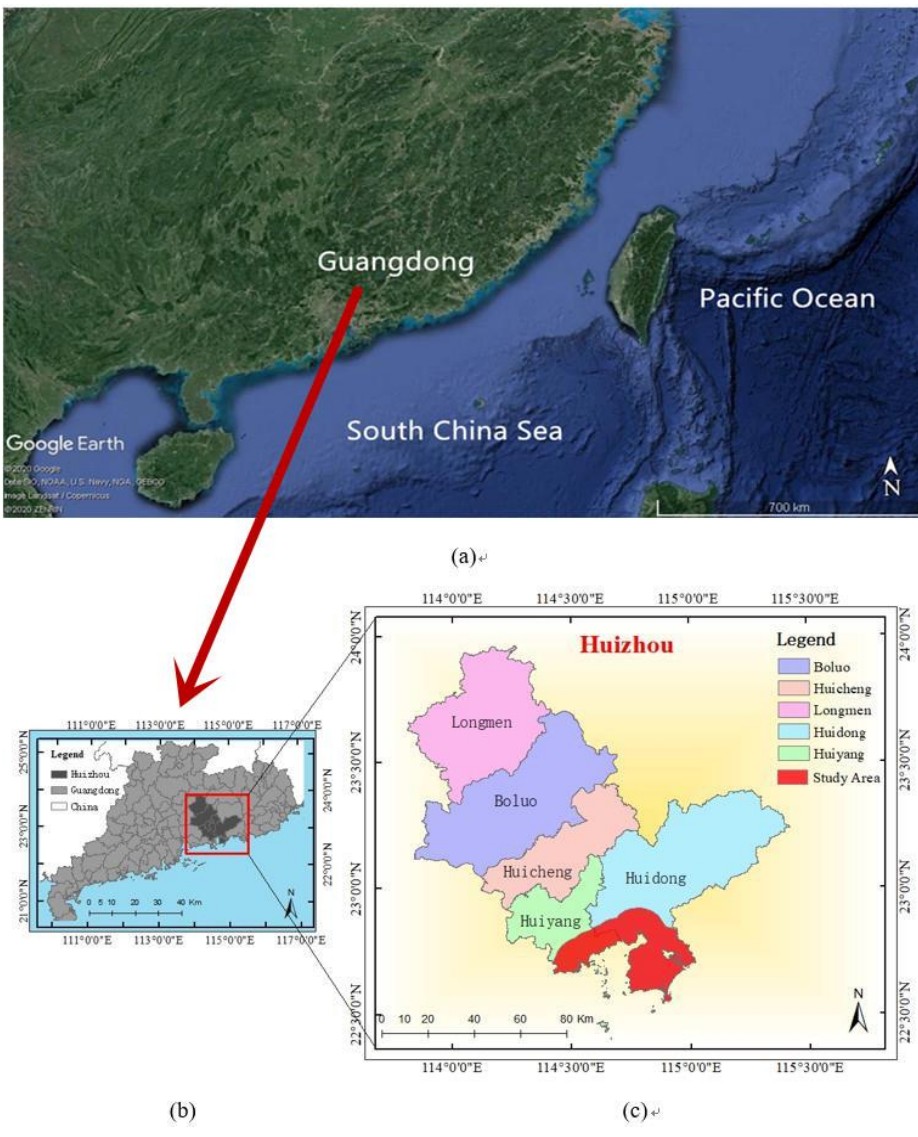

**Figure 1.** The maps of observation locations used in the study: (a) The satellite image displays the Guangdong province in the southern area of China, which was obtained from the Google Earth (Map data ©2020 Google, © 2020 ZENRIN); (b) The map of Guangdong Province where the shapefile of administrative boundaries was obtained from GADM database under ©2018 GADM license (https://gadm.org/data.html), with academic publishing permission from Global Administrative Areas (https://gadm.org/license.html); (c) The map of Huizhou city and the study area, which was made with ArcGIS 10.5. The maps and satellite images obtained from the Google Earth or the Google Maps can be used and printed in the research papers with permission from google website (https://www.google.com/permissions/geoguidelines/).

In this paper, the region within a distance of 10 kilometers from the coastline in Huiyang district and Huidong county is chosen as a study area to understand the potential risk of storm surge in this region, as shown in Figure 1 (c). In addition to



high population density in coastal communities, the main reason for choosing this region is that high concentration of petroleum facilities and infrastructures in the Daya Bay Petrochemical Zone make the study area vulnerable to storm surges. The Daya Bay Petrochemical Zone has an area of about 27.8 km$^2$ and is currently taking the first spot at the scale of

petrochemical-refining integration in China. By the end of 2018, the Petrochemical Zone has been commenced business by many world's top 500 companies and industry-leading enterprises. These world chemical and industrial giants including Exxon Mobil, Shell, and Clariant have invested 131.6 billion yuan to shape up the industrial chains of the oil refinery, ethylene, propylene, and butylene in Daya Bay Petrochemical Zone. In 2018, the oil refining capacity and the ethylene production capacity have been enhanced to 22 million tons/year and 2.2 million tons/year, respectively, and the

petrochemical industrial output value reached 270 billion yuan. Now, the Daya Bay Petrochemical Zone is striving to develop into the world-class petrochemical base and planning to be one of the world's top ten petrochemical industrial zones in the subsequent few years. Therefore, with the growing population density and particularly the rapid development of petroleum and chemical industries, the risks and vulnerability in the study area to storm surge will increase. The risk assessment and risk analysis are considered to be important strategies to identify the risk regions in the Daya Bay

Petrochemical Zone, which can minimize the loss of life and property and prevent environmental damage caused by affected coastal petroleum facilities and infrastructures.

## 2.2 Datasets requirement

The datasets used in the paper contain observed data and survey data obtained from various sources. The datasets can be employed to conduct the hazard assessment, vulnerability assessment, and risk assessment of storm surge in the study area

for each of the different typhoon intensity scenarios. The datasets are listed in Table 1 and described below.

**Table 1.** Summary of the datasets used in risk assessment of storm surge.

| Dataset Name | Source | Time |
|---|---|---|
| Historical Tropical Cyclone | China Meteorological Administration | 1949-2017 |
| Digital Elevation Model (DEM) | Huizhou Land and Resources Bureau | 2015 |
| Storm Surge Barriers | Huizhou Oceanic Administration | 2018 |
| Land Cover Types | Huizhou Land and Resources Bureau | 2016 |
| District Boundaries | Huizhou Land and Resources Bureau | 2017 |
| Water Level Records | Huizhou tidal gauge station | 2006-2018 |

1). The dataset of Historical Tropical Cyclone. It contains information on name, time, track, minimum central pressure,

and maximum wind velocity of tropical cyclones that made landfall in the coastal area of Huizhou during the period from 1949 to 2017. It was obtained from the China Meteorological Administration. The historical data can be analysed to





construct representative typhoon scenarios over the study area. The input parameters of each of typhoon scenarios are used to generate wind field with Jelesnianski method, which is the requirement for modelling storm surge.

    2). The dataset of Digital Elevation Models (DEM). The dataset of DEM with a scale of 1:2000 was constructed in 2015

and is available from Huizhou Land and Resources Bureau. It is a raster dataset that depicts land heights in Huizhou's solid surface. The point in the dataset contains the elevation value for the region that point covers. The DEM dataset can be used in modelling storm surge.

    3). The dataset of Storm Surge Barriers. The dataset includes information on storm surge barriers including dikes and levees constructed in the study area, as shown in Figure 2. The information such as name, height, slope, location, material,

latitude, and longitude, which have been surveyed in the project named marine disaster prevention and reduction of Huizhou in 2018. The dataset can be utilized to accurately simulate the inundation extent and depth of storm surge.

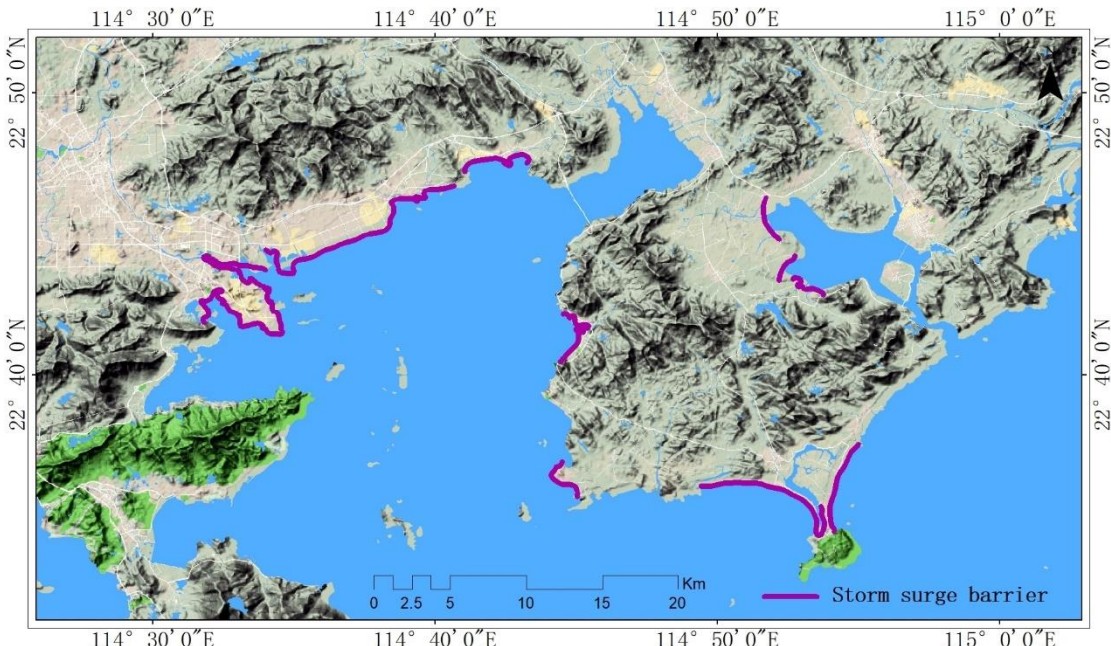

**Figure 2.** The map of the study area including the Storm Surge Barriers such as dikes and levees along the coastline of Huizhou, which are indicated by the violet lines. The map was made with ArcGIS 10.5 software based on the terrain base map layer, which was obtained from

Google Maps (Map data ©2019 Google).

    4). The dataset of Land Cover Types. The dataset contains current physical material including water, agriculture, tree, and wetlands over the study area. It was obtained from the Huizhou Land and Resources Bureau, which was created by analyzing remotely sensed imagery of Huizhou in 2016. Because each physical material has its bottom friction, the potential inundated

area of modelling storm surge cannot be the same on the different combinations of land covers. The dataset can be applied to vulnerability assessment.



5). The dataset of Administrative Boundary. The dataset contains administrative boundaries at the township level of Huizhou in 2017 and there are 12 towns in the study area. It was obtained from Huizhou Land and Resources Bureau.

6). The dataset of Water Level Records. The Huizhou Oceanic Administration deployed the Huizhou water level gauging station, which is located in the Quenwan port, in 2006, as shown in Figure 3. The coastal water level around the Quenwan port is automatically measured and recorded with a gauge at fixed intervals of time. The dataset contains the height of hourly water level records during the period from 2006 to 2018. The dataset can be used for validating the coupled model (ADCIRC–SWAN) by comparing simulated water levels and measured water levels.

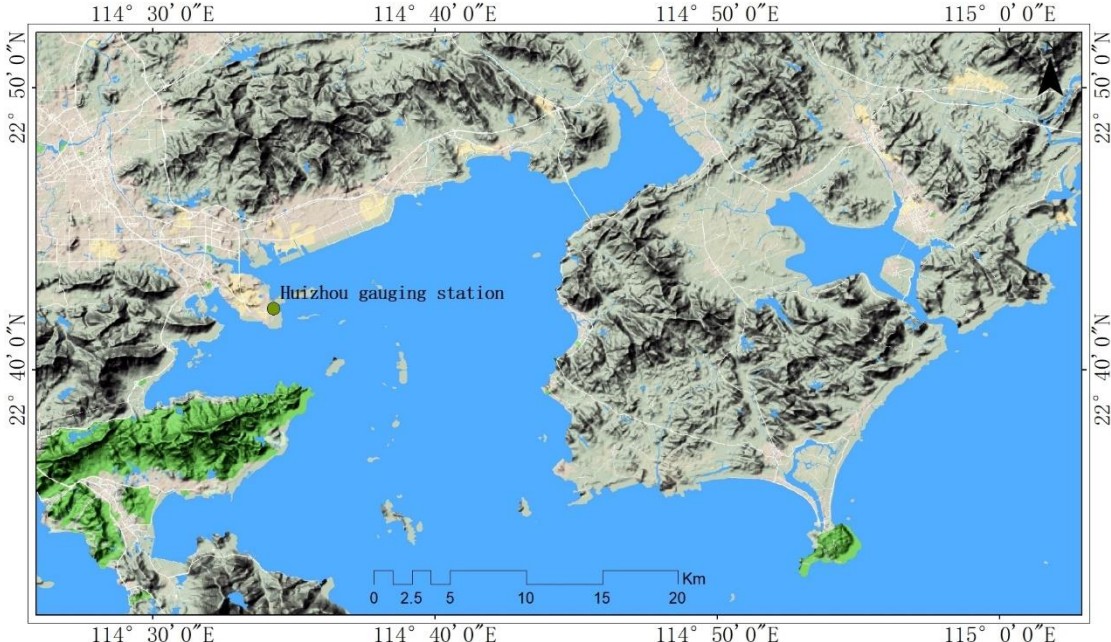

**Figure 3.** The map of the study area including the Huizhou gauging station in the Quenwan port, which is indicated by the green dot. The map was made with ArcGIS 10.5 software based on the terrain base map layer, which was obtained from Google Maps (Map data ©2019 Google).

## 3 Methodology and procedure

### 3.1 Model description and validation

In this study, the Jelesnianski numerical method (Jelesnianski and Taylor, 1973), the well-established Advanced Circulation (ADCIRC) model (Luettich et al., 1992, Westerink et al., 1994) and Simulating Waves Nearshore (SWAN) model (Booij et al., 1999) are employed to simulate tropical storm surge. The ADCIRC model is a two-and three-dimensional hydrodynamic circulation model, which can be utilized to simulate wind-driven circulation and provides water elevation and velocity of the current in coastal seas. The model solves the continuity and the momentum equations for moving fluid and water elevation.





It has been applied to simulate the hydrology in regions including the Gulf of Mexico, the Mediterranean Sea, and the South China Sea. The SWAN model is a third-generation numerical wave model, which is used to simulate wind-generated wave propagation in coastal regions. The model computes the wave action density spectrum by solving the wave action balance equation. It can be coupled to the ADCIRC model to simulate the storm surge on the same unstructured grid (Dietrich et al., 2011; Dietrich et al., 2012).

The computational domain in this study covered the coastal region of Huizhou, as shown in Figure 4, with progressively higher resolution approaching the coastal area of Huizhou. In the region along the coastline of Huizhou, the high grid resolution of 100-200 m is provided to improve the simulation accuracy, and simultaneously unstructured grids contain a coarse resolution of 30 km along the open ocean boundary for decreasing computational cost. There are 38407 nodes and 74328 grids over the computational domain, as shown in Figure 4 (a). The open boundary of the model water level is

controlled by the total water level, which is obtained by the superposition of 11 astronomical tidal components. The wave open boundary is controlled by a two-dimensional wave spectrum.

The procedure for modelling storm surge is as follows: the wind filed, which is generated by the Jelesnianski method, is provided to the coupled model (ADCIRC-SWAN). Then, the ADCIRC model is operated to calculate the water level and current with the wind filed. Subsequently, based on the water level, the current, and the wind velocity, the SWAN model

computes the wave spectrum, which is then passed back to the ADCIRC model to calculate the water levels in the next simulated round. Thus, the modelling typhoon event can be converted by the wave-current coupled model (ADCIRC-SWAN) into a storm surge event, outputting the data of water height corresponding to every grid node over the computational domain.




(a)

(b)

(c)

**Figure 4.** A computational domain (a) and the domain over the study area (b) (c). The satellite image (c) was obtained from the Google Earth (Map data ©2020 Google, © 2020 ZENRIN).

As the Jelesnianski method and the coupled model (ADCIRC-SWAN) have never been used in modeling storm surges for the coastal areas of Huizhou, the simulated performance of the coupled model is needed to be evaluated. The 10 representative typhoons (0812、0814、0906、1208、1319、1604、1622、1713、1720、1822), which caused higher water levels in Huizhou gauging station (Figure 3), were selected to validate the coupled model (ADCIRC–SWAN) for the study area. Figure 5 shows all maximum simulated water levels, the highest observed water levels from the 10 representative typhoons, and the timing of these peaks for these 10 representative typhoons.

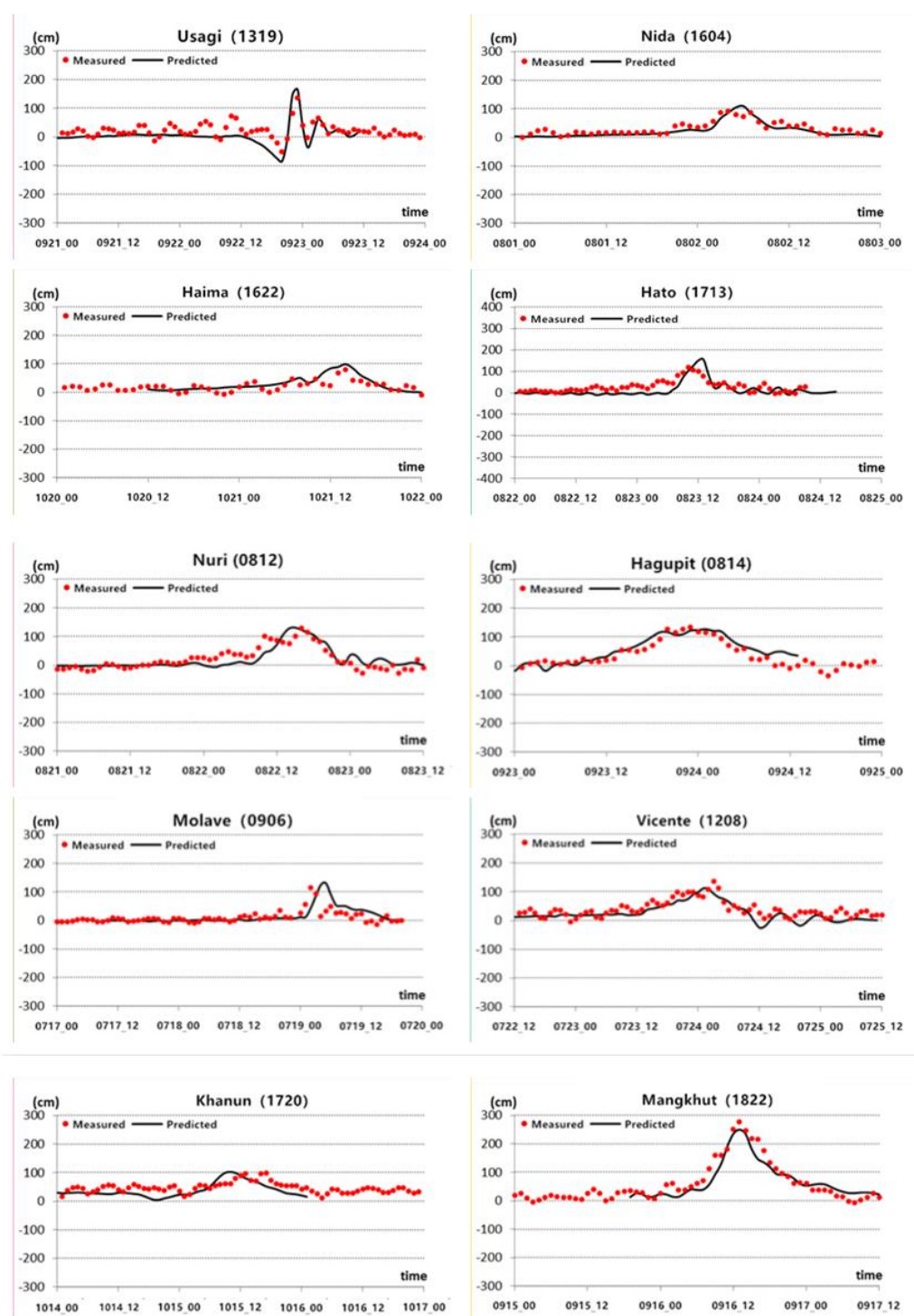

**Figure 5.** The predicted water levels (in black line) and highest measured water levels (in red dots) recorded by the Huizhou gauging station in the Quenwan port during the typhoon period.





Two error statistic approaches were utilized to evaluate the performance of the coupled model with a comparison between maximum predicted water levels and the highest measured water levels. The Absolute Error (AE) is computed when the highest measured water level is above 100 cm. The Relative Error (RE) is calculated as the measured observed water level is

below 100 cm. The statistical result, as summarized in Table 2, indicates that the data with RE≤20% or AE≤20 cm account for 80% of all simulated data, which satisfy the criterion in the *guideline*. Therefore, the predicted performance of the coupled model is considered to be reliable regarding its ability to simulate storm surges in the study area.

**Table 2.** Summary of error statistics (AE and RE) between maximum predicted water levels and highest measured water levels

| Typhoon Name | Date | Measured Data(cm) | Relative Error(%) | Absolute Error(cm) |
|---|---|---|---|---|
| Nuri (0812) | 21 August 2008 | 129 | 2 | / |
| Hagupit (0814) | 23 September 2008 | 135 | 6 | / |
| Molave (0906) | 17 July 2009 | 116 | 13 | / |
| Vicente (1208) | 22 July 2012 | 136 | 18 | / |
| Usagi (1319) | 21 September 2012 | 137 | 21 | / |
| Nida (1604) | 01 August 2016 | 92 | / | 18 |
| Haima (1622) | 20 October 2016 | 81 | / | 18 |
| Hato (1713) | 22 August 2017 | 120 | 30 | / |
| Khanun (1720) | 14 October 2017 | 98 | / | 5 |
| Mangkhut (1822) | 15 September 2018 | 278 | 10 | / |

**3.2 Storm parameters**

After validating the coupled ADCIRC-SWAN model, the input storm parameters are needed to be set for designing the typhoon scenarios, which can be used to create the wind field with the Jelesnianski method to drive the storm surge in the coupled model. The storm parameters include the minimum central pressure, the radius of maximum winds, the maximum wind velocity, and the storm track.

**3.2.1 Return period and minimum central pressure**

The typhoon return period is the average time between typhoons with a certain intensity at a specific location. The typhoon return periods for Huizhou can be measured through the dataset of Historical Tropical Cyclone. After analyzing the minimum central pressure records from the historical dataset, the minimum central pressures-return period relation for the study area was constructed with extreme value distribution Type-I, as shown in Figure 6. Thus, the minimum central

pressure associated with a given return period can be calculated. For example, from Figure 6, the minimum central pressure

of typhoons consisted with the return period of 1000-year was estimated at 880 hPa.

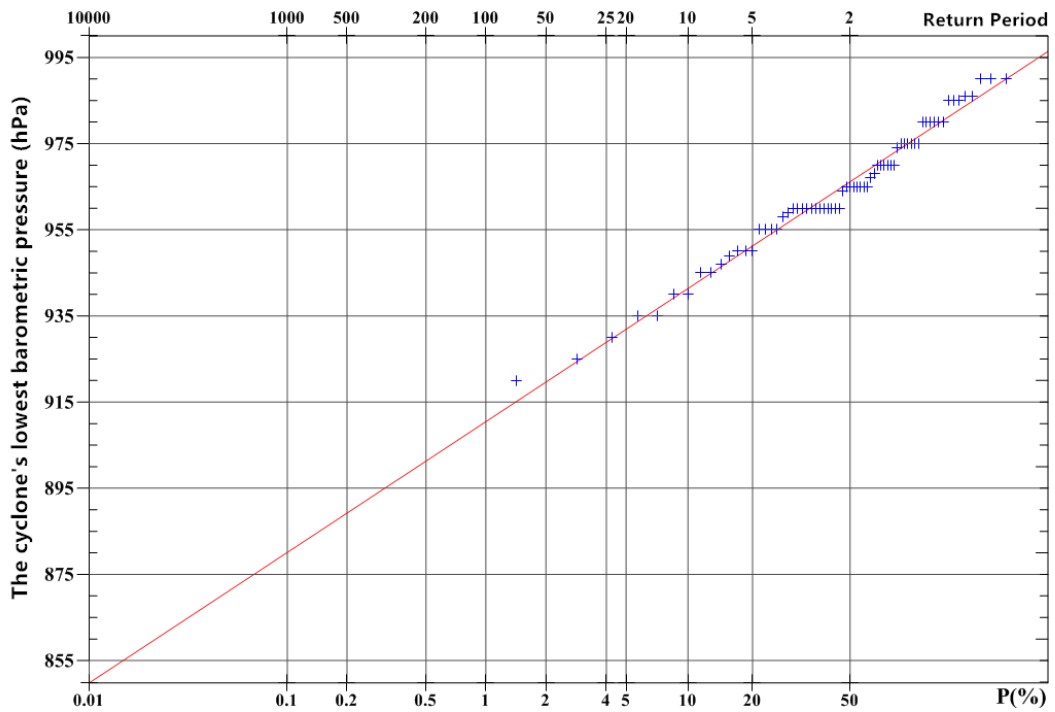

**Figure 6.** The relation between the return period and the central pressure of typhoons on extreme value distribution Type-I, which struck

coastal areas of Huizhou city from 1949-2017.

**3.2.2 Radius of maximum wind**

The radius of maximum wind ($R_{max}$) has a strong connection to the maximum wind velocity and the minimum central

pressure of the typhoon ($P_0$). The researchers have developed empirical formulas to calculate the radius of maximum wind

based on the $R_{max}$–$P_0$ relationship, as shown in Eq. from (1) to (3) where $\emptyset$ represents the latitude of the typhoon's center, $\Delta P$

indicates the pressure difference between central pressure and ambient pressure, $R_k$ is empirical constant with the value of 50.

$$R_{max} = \exp(2.635 - 0.00005086\Delta P^2 + 0.0394899\emptyset) \tag{1}$$

$$R_{max} = 1119.0 \times (1010 - P_0)^{-0.806} \tag{2}$$

$$R_{max} = R_k - 0.4 \times (P_0 - 900) + 0.01 \times (P_0 - 900)^2 \tag{3}$$

Collecting the historical radius of maximum wind data measured in the northwest Pacific hurricane records (2001-2016)

from the Joint Typhoon Warning Center, the empirical Eq. (4) of the $R_{max}$–$\Delta P$ relation can be constructed based on Eq. (2)

with regression statistics, as shown in Figure 7.

$$R_{max} = 260.93 \times (1010 - P_0)^{-0.512} \tag{4}$$

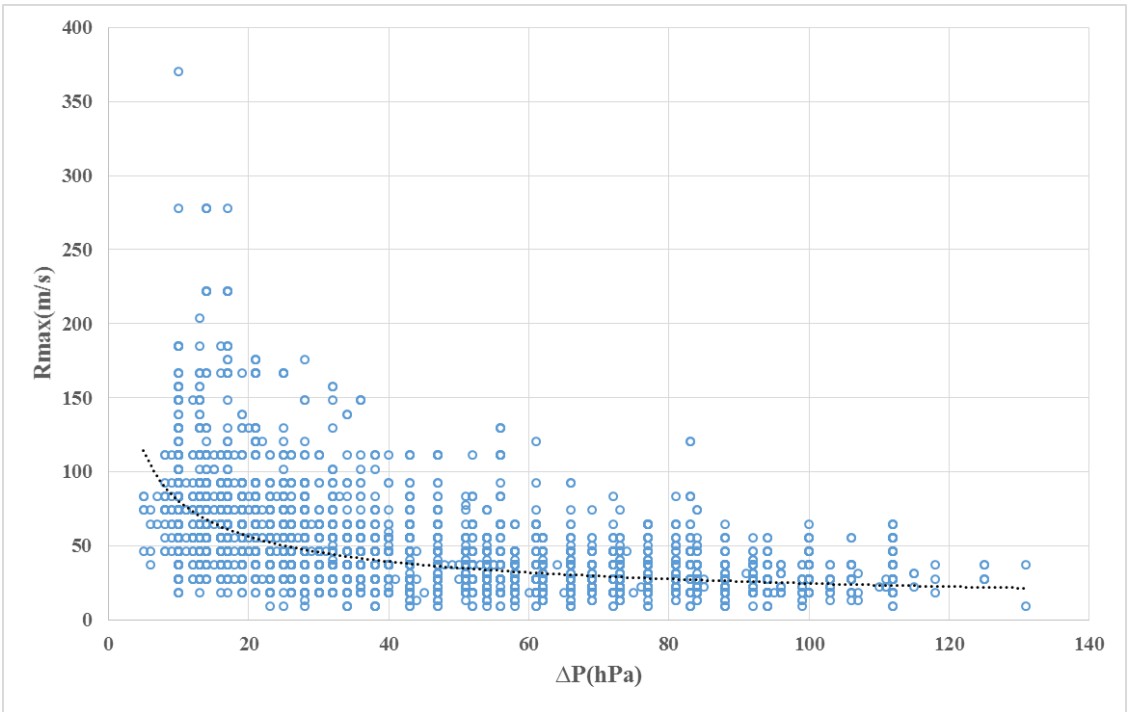

**Figure 7.** The relation between the pressure difference ($\Delta P$) and the radius of maximum wind ($R_{max}$)

Therefore, according to the above empirical equations and historical observations, the radius of maximum wind consisted
with the minimum central pressure can be calculated. As shown in Table 3, averaging the empirical values from Eq. (1) to (4)
and observed values, the radius of maximum wind with respect to the minimum central pressure at 880 hPa was estimated at
30 km.

**Table 3.** The estimated value of the radius of maximum wind when $P_0$ is 880 hPa

| | Empirical Eq. | | | | Observed | Averaged |
|---|---|---|---|---|---|---|
| | (1) | (2) | (3) | (4) | JTWC | |
| **Radius(km)** | 15 | 22 | 62 | 22 | [9,37] | 30 |

### 3.2.3 Maximum wind velocity

According to the records from the dataset of Historical Tropical Cyclones, the regression statistics method is applied to
processing the observed data including maximum wind velocity ($V_{max}$) and the minimum central pressure ($P_0$) with
the wind-pressure empirical equation (Atkinson and Holliday, 1977), as shown in Figure 8.


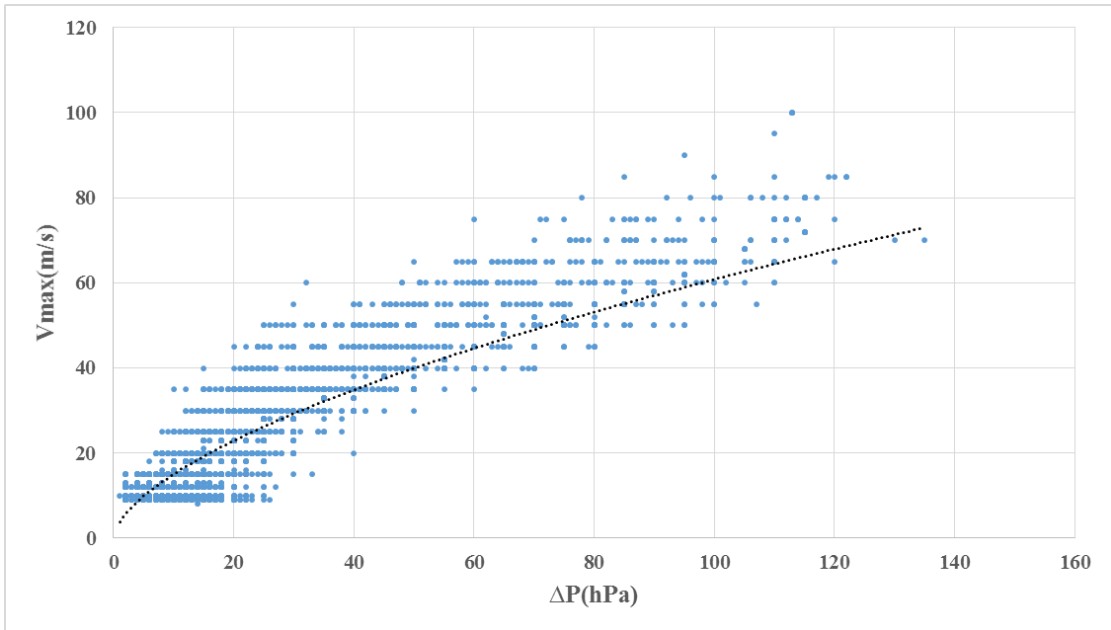

**Figure 8.** The relation between the pressure difference ($\Delta P$) and the maximum wind velocity ($V_{max}$)

Then, the empirical equation of the $V_{max}$–$P_0$ over the study area can be established, as shown in Eq. from (5).

$$V_{max} = 3.7237 \times (1010 - P_0)^{0.6065} \tag{5}$$

### 3.2.4 Tropical storm track

The tracks of tropical cyclones, which that affected the coastal area of Huizhou during the period from 1949 to 2017, can be divided into four categories: moving northward (10.2% of total), moving northwestward (47.5% of total), moving west-southwestward (15.5% of total), and moving northeastward (24.4% of total), as shown in Figure 9.


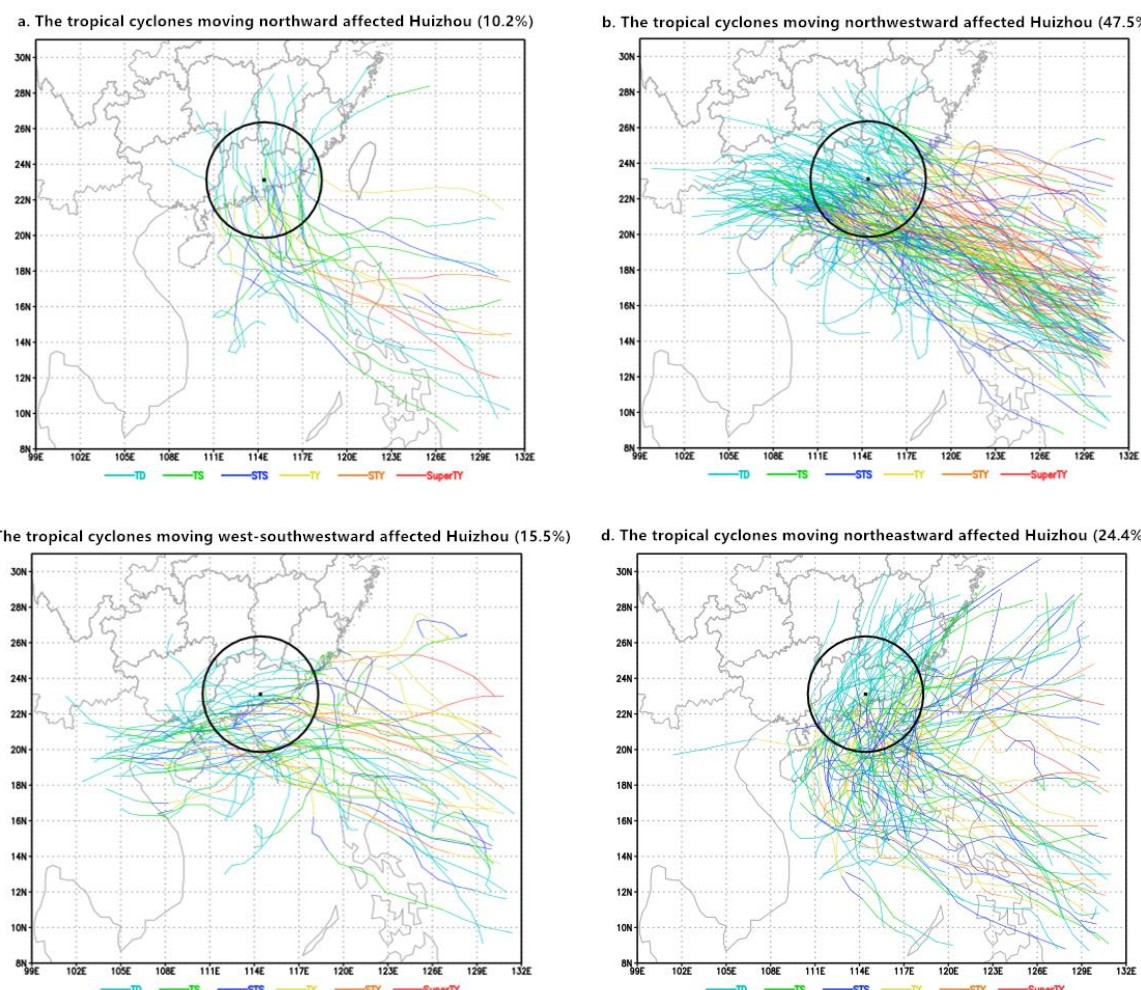


**Figure 9.** The tracks of tropical cyclones affected Huizhou from 1949 to 2017.

Figure 9 indicates that tropical cyclones approached the coast areas of Huizhou on the northwest (47.5%) are the most
common. The northwestward track is perpendicular to the coastline of Huizhou, which can increase the maximum storm
surge for the study area. Analyzing the dataset of Historical Tropical Cyclones and dataset of Water Level Records, the five
typhoons (7908、0906、1319、1713、1822) that caused the higher water level in the study area were moved on the
northwestward track. Especially, the recent super typhoon Mangkhut (1822), which struck southern China in September
2018, led to the highest water level in Huizhou history. The tract of super typhoon Mangkhut is shown in Figure 10.





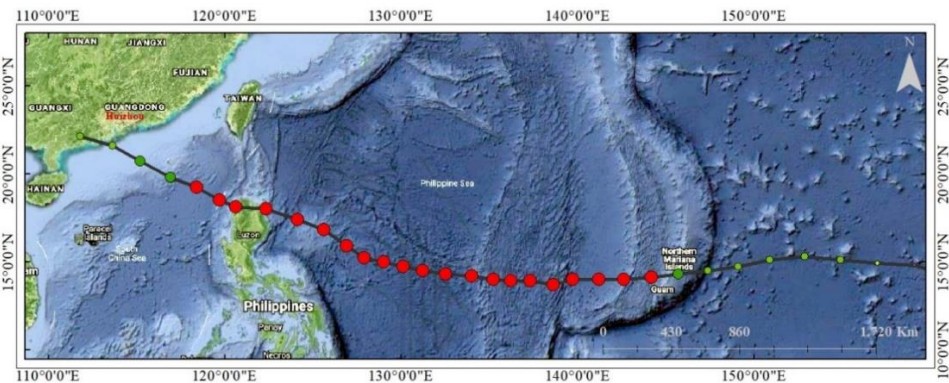

**Figure 10.** The track of super typhoon Mangkhut over the western North Pacific and the South China Sea. The track map was made with ArcGIS 10.5 software based on the satellite base map layer, which was obtained from Google Maps (Map data ©2019 Google).

Figure 10 depicts the super typhoon Mangkhut moving northwestward into the South China Sea toward southern China. It

has been the costliest and devastating marine disaster in the history of Huizhou. During the super typhoon Mangkhut period, the maximum water level at Huizhou gauging station reached up to 349 cm at 13:00, September 16, 2018. The direct economic losses caused by super typhoon Mangkhut to Huizhou was estimated at 577.39 million RMB, and 0.236 million people were affected. Therefore, the Mangkhut track was chosen to model the worst–case scenario of storm surge under different representative typhoon intensities in this study.

In order to provide complete geographical coverage for the study area, a set of deviated typhoon tracks were produced. The 33 typhoon tracks deviating from the original Typhoon Mangkhut track were generated, as shown in Figure 11. These 34 tracks are spaced 5 km apart and represent the typhoon activity near Huizhou. In this paper, the tracks were used to simulate the storm surges in the coastal area of Huizhou with the Jelesnianski method and the coupled model (ADCIRC–SWAN).

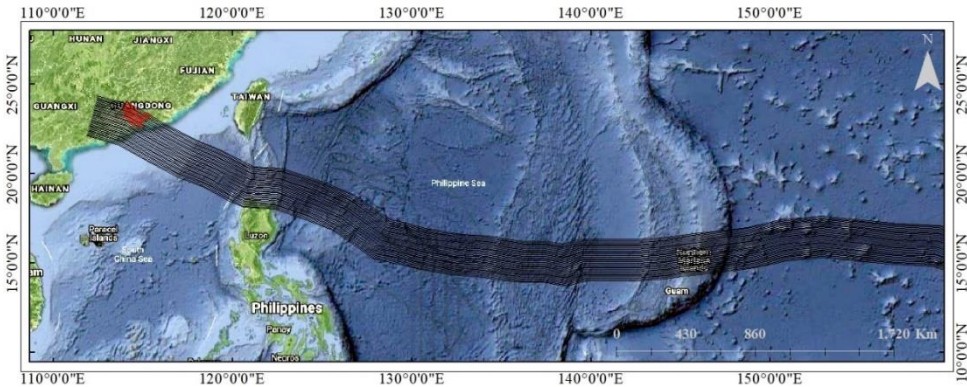


**Figure 11.** The 34 typhoon tracks (16 left-side shifting and 17 right-side shifting from the Mangkhut track) over the western North Pacific and the South China Sea. The track map was made with ArcGIS 10.5 software based on the satellite base map layer, which was obtained from Google Maps (Map data ©2019 Google).




### 3.3 Procedure for risk assessment

The standard operating procedure for risk assessment of storm surge in China is derived from the *standard Technical guideline for risk assessment and zoning of marine disaster (Part 1: storm surge)*. The risk is regarded as a function of hazard and vulnerability. Thus, the procedure includes five steps: data collection, typhoon scenarios design, hazard assessment, vulnerability assessment, and risk assessment as shown in Figure 12.

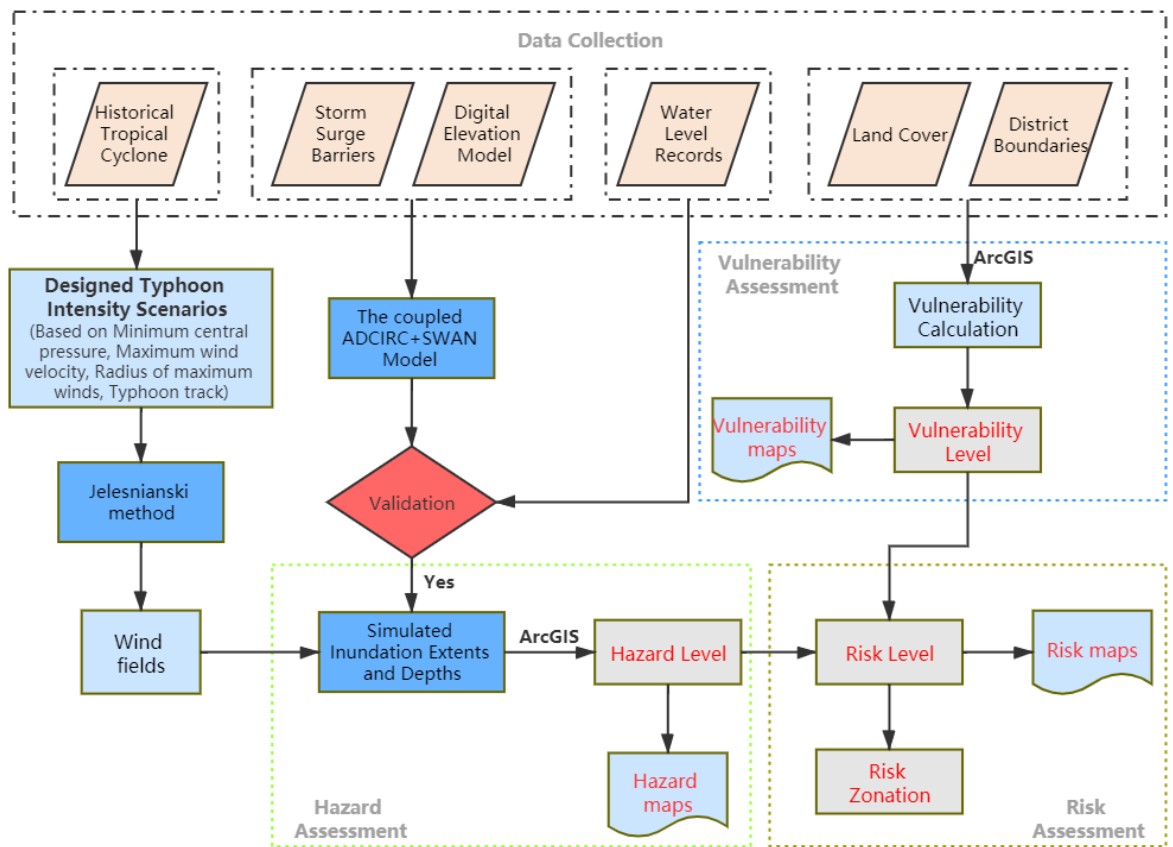

**Figure 12.** The procedure for risk assessment of storm surge in the study.

As seen in Figure 12, the input data includes Land Cover, DEM, Historical Tropical Cyclones, Historical Sea Level, Storm Surge Barriers, and District Boundaries. The wind field created with the Jelesnianski method is provided to the coupled (SWAN+ADCIRC) model, which simulates the storm surge for each of the design typhoon intensity scenarios.

Subsequently, the 12-hour time series data of simulated surge documented the temporal variation of inundation depth over the computational domain are generated. Then, these output data from the coupled (SWAN+ADCIRC) model for each of the scenarios are converted to ArcGIS 10.5 software. Eventually, the hazard maps, vulnerability maps, and risk maps for these representative scenarios are made in the ArcGIS 10.5 software.





### 3.3.1 Typhoon scenario design

The typhoon scenario is parametrized by intensity, maximum wind velocity, radius of maximum winds, and track. The lower the central pressure or the longer the year return period, the more intense the storm. Thus, the minimum central pressure or the year return period can be regarded as an indicator for the typhoon intensity. The comprehensive and representative typhoon intensity for the minimum central pressure of 880 hPa, 910 hPa, 920 hPa, 930 hPa, and 940 hPa (corresponding to 1000-year, 100-year, 50-year, 20-year, and 10-year return period) scenarios were designed. The

corresponding maximum wind velocity and radius of maximum winds were calculated for each of the designed scenarios according to the analysis in section 3.2.1- section 3.2.3, as shown in Table 4. The 34 constructed tracks discussed in section 3.2.4 were used to create the wind field.

**Table 4.** Constructed storm scenarios with different intensities.

| Grade | I | II | III | IV | V |
|---|---|---|---|---|---|
| Minimum central pressure (hPa) | 880 | 910 | 920 | 930 | 940 |
| Return period (year) | 1000 | 100 | 50 | 20 | 10 |
| Maximum wind velocity (m/s) | 61 | 57 | 53 | 49 | 45 |
| Radius of maximum winds (km) | 30 | 31 | 33 | 35 | 38 |
| The number of typhoon track | 34 | 34 | 34 | 34 | 34 |

### 3.3.2 Hazard assessment

The hazard assessment is to identify the potential inundation extent and depth of storm surge caused by each of the designed typhoons for the study area. The ADCIRC+SWAN model integrated with the Jelesnianski method was run on the datasets for scenarios with an increasing minimum central pressure from 880 hPa up to 940 hPa. Then, the outputs of the coupled model (ADCIRC–SWAN) were imported to the GIS software.

The spatial extents of surge area and heights of surge water in given scenarios were displayed in the ArcGIS 10.5 software. The inundation depth was calculated by subtracting DEM from the height of simulated surge water at each grid. The storm surge hazard for the study area was assessed based on the classifications of inundation depths as summarized in Table 5. Accordingly, the different hazard levels was assigned to the inundation zones.






**Table 5.** Hazard zone classification according to the inundation depth.

| Inundation Depth (cm) | Hazard |
|---|---|
| [15, 50) | Low (IV) |
| [50, 120) | Moderate (III) |
| [120, 300) | High (II) |
| [300, +∞) | Very High (I) |

### 3.3.3 Vulnerability assessment

The exposure assessment aims at identifying elements affected by storm surge. The land cover can be seen as the representation of affected elements. The land cover type is regarded as an indicator to assess the vulnerability in the study area to storm surge.

The vulnerability values ranging from 0 to 1 are assigned to different land cover types, which were defined in the *guideline* according to their properties of susceptibility and resilience to storm surge. The value of 0 indicates no vulnerability and the value of 1 represents the highest vulnerability. The four levels of vulnerability were defined in the *guideline* (I, II, III and IV) from very high vulnerability (I) to low vulnerability (IV). The Land Cover dataset was categorized into 12 first classifications according to the *guideline*, as summarized in Table 6. Based on the vulnerability value corresponding to land cover type in Table 6, the vulnerability level over the study area was evaluated.

**Table 6.** The vulnerability value and vulnerability level for different land cover types.

| The first classification of land cover | | | The second classification of land cover | | | |
|---|---|---|---|---|---|---|
| Number | Name | Value | Number | Name | Value | Level |
| 01 | Agriculture | 0.1~0.2 | 011 | Paddy field | 0.1 | IV |
| | | | 012 | Irrigable land | 0.2 | IV |
| | | | 013 | Dry land | 0.2 | IV |
| 02 | Garden plot | 0.1~0.3 | 021 | Orchard | 0.3 | IV |
| | | | 023 | Other filed | 0.1 | IV |
| 03 | Forest | 0.1 | 031 | Forest land | 0.1 | IV |
| | | | 032 | Shrubland | 0.1 | IV |
| | | | 033 | Other woodland | 0.1 | IV |





| 04 | Pasture | 0.1 | 043 | Other Grassland | 0.1 | IV |
| 06 | Mining storage | 0.6~1 | 062 | Land for mining | 0.6~0.9 | II~I |
| 07 | Settlements | 1 | 071 | Urban residential land | 1 | I |
| | | | 072 | Rural residence land | 1 | I |
| 08 | Urban infrastructure | 0.4~1 | 088 | Scenic area | 0.5 | III |
| 10 | Transportation | 0.6~1 | 101 | Land for railway | 0.6~0.9 | II~I |
| | | | 102 | Land for highways | 0.6~0.8 | II |
| | | | 104 | Country road | 0.6 | II |
| | | | 105 | Land for airport | 0.8~1 | II~I |
| | | | 106 | Land for harbour | 0.6~1 | II~I |
| | | | 107 | Land for pipeline transportation | 0.6~1 | II~I |
| 11 | Water | 0.1~0.8 | 111 | River | 0.1 | IV |
| | | | 113 | Reservoir | 0.2 | IV |
| | | | 114 | Pond | 0.3 | IV |
| | | | 115 | Coastal beach | 0.1 | IV |
| | | | 116 | Inland beach | 0.1 | IV |
| | | | 117 | Ditch | 0.1 | IV |
| | | | 118 | Hydraulic construction land | 0.5~0.8 | III~II |
| 12 | Others | 0.1~0.5 | 122 | Facility agriculture land | 0.2~0.5 | IV~III |
| | | | 124 | Saline-alkali land | 0.1 | IV |
| | | | 125 | Wetland | 0.1 | IV |
| | | | 126 | Sand land | 0.1 | IV |
| | | | 127 | Bare land | 0.1 | IV |

### 3.3.4 Risk assessment

In the process of the risk assessment, the inundated region is divided into several storm surge risk districts by integrating the inundation hazard assessment and vulnerability of affected elements in the study area. The quantitative risk assessment and the risk matrix are the primary methods in risk assessment. However, the quantitative risk assessment method is data





demanding, and it is difficult to quantify all population and property at risk. The risk matrix, a typical semi-quantitative approach, is utilized to solve these problems. The risk matrix is made of classes of hazard level on one axis and
the vulnerability level on the other axis, as shown in two-dimensional Table 7.

**Table 7.** Risk matrix method for assessing storm surge risk.

|  |  | Vulnerability | | | |
|---|---|---|---|---|---|
|  |  | Low (IV) [0.1,0.3] | Moderate (III) (0.3,0.5] | High (II) (0.5,0.8] | Very High (I) (0.8,1] |
| **Hazard** | Low (IV) | Low Risk (IV) | Low Risk (IV) | Moderate Risk (III) | Moderate Risk (III) |
|  | Moderate (III) | Low Risk (IV) | Moderate Risk (III) | High Risk (II) | Very High Risk (I) |
|  | High (II) | Moderate Risk (III) | High Risk (II) | High Risk (II) | Very High Risk (I) |
|  | Very High (I) | Moderate Risk (III) | High Risk (II) | Very High Risk (I) | Very High Risk (I) |

With the risk matrix approach, the degree of risk can be determined based on limited quantitative data. The degree of risk
is evaluated by four levels (I, II, III, and IV) from very high risk (I) to low risk (IV). For example, low vulnerability combined with low hazard can lead to a low risk, or the combination of very high vulnerability and low hazard can lead to moderate risk in the area.

## 4 Results and discussion

### 4.1 Hazard assessment

The coupled model (ADCIRC–SWAN) model and the Jelesnianski method were utilized to simulate the inundation extents and depths for each of the designed typhoon scenarios (Table 4). The 12-hour simulation of storm surge flooding over the study area for each of typhoon scenarios can be displayed in the ArcGIS 10.5 software. For example, for the typhoon intensity with the 880 hPa (1000-year return period), the simulated inundation depths and extents of storm surge at time intervals of 2 hours during the 11-hour period are shown in Figure 13. The inundation depth over the study area was divided
into four categories according to the criterion in the *guideline*.

**Figure 13.** The maps display that the simulated inundation extents of storm surge over the study area during 12-hour simulation for the 880 hPa (1000-year return period) scenario. These maps were made with ArcGIS 10.5 software based on the terrain base map layer, which was obtained from Google Maps (Map data ©2019 Google).

As shown in Figure 13, the inundation area progressively expanded from coastline to the mainland and reached a maximum at approximately the 11th hour. Moreover, the maximum distance that storm surge flooding penetrates inland from the coastline is approximately 6 km and the inundation distances in other regions are less than 4 km. Furthermore, the predicted inundation depths over most of the coastal areas are more than 300 cm at the 11th hour.

With the output data of simulated storm surge elevations at the 12th hour, the hazard map where regions are in different colors based on the inundation depths (Table 5) was made for each of the designed scenarios in the ArcGIS 10.5 software. The higher the inundation depth reach, the higher the risk is. These maps of hazard assessment over the study area for the representative scenarios are shown in Figure 14.


(a)

(b)                                    (c)

(d)                                    (e)

**Figure 14.** The hazard assessment maps represent the inundation extents and depths for five storm scenarios ((a):880 hPa/1000-year,

(b):910 hPa/100-year, (c):920 hPa/50-year, (d):930 hPa/20-year, (e):940 hPa/10-year). Different colors represent different hazard levels.

The red means the highest water level and very high storm surge hazard, orange stands for high hazard, yellow represents moderate hazard,

and blue indicates low hazard on the map. The important locations in the study area are labeled with letters and the violet lines refer to the

storm surge barriers along the coastline of Huizhou. These maps were made with ArcGIS 10.5 software based on the terrain base map

layer, which was obtained from Google Maps (Map data ©2019 Google).





Figure 14 shows the ultimate simulated inundation extents and corresponding hazard assessment under each of the designed scenarios. First of all, the maximum inundation depths and extents decrease with increasing typhoon intensity. Second, in terms of the Daya Bay Petrochemical Zone (B), the hazard is in high or very high level to storm surges when the

minimum central pressure of typhoon is 880 hPa. Moreover, Daya Bay Petrochemical Zone (B) has moderate and high hazard levels to storm surge as the minimum central pressure of typhoon is 910 hPa or 920 hPa. Furthermore, the Daya Bay Petrochemical Zone (B) is hardly affected by the storm surge from the typhoon with a minimum central pressure at 930 hPa or 940 hPa due to the protection given by storm surge barriers.

The Figure 14 (a) demonstrates that the very high hazard zones are mainly located in the southeast of the Daya Bay

Development Zone (D), the Huizhou Port (C), the region to the north of the White Sand Shore (A), the field close to the Huidong Station (F) in Renshan Town (G), the coastal place from the Double-Moon Bay (I) to the Pinghai Town (H), the Tieyong Town (N), the land from the Huangbu Town (K) to Yandunling (M), and Yanzhou Peninsula (L). The construction of storm surge barriers along the coastline of Huizhou cannot prevent storm surge caused by typhoon with a minimum central pressure at 880 hPa.

Figure 14 (b)-(e) indicate that the typhoons with minimum central pressure are at 910 hPa, 920 hPa, 930 hPa, and 940 hPa, most areas of the inundated region are under moderate level or high hazard level. The storm surge barriers can effectively protect coastal urban settlements and industrial facilities from storm surge generated by these designed typhoon scenarios. However, the Huizhou Port (C) and the southeast of the Daya Bay Development Zone (D) are under a very high hazard level caused by storm surge as they lie along the low elevation coastal area. In addition, the region to the north of the White Sand

Shore (A), the southwest of the Gilong town (J), and the Yanzhou Peninsula (L) are in very high hazard to the storm surge because of relatively flat without protection from barriers.

**4.2 Vulnerability assessment**

The vulnerability assessment can be used to identify different vulnerable regions in the study area. Making an exposure map is an important step before the vulnerability assessment. The Land Cover dataset obtained from Huizhou Land and

Resources Bureau in 2016 can be categorized into 10 different land cover types and the exposure map over the study area was illustrated in Figure 15. In addition, the land areas and percentages of different land cover types were presented in Figure 16.


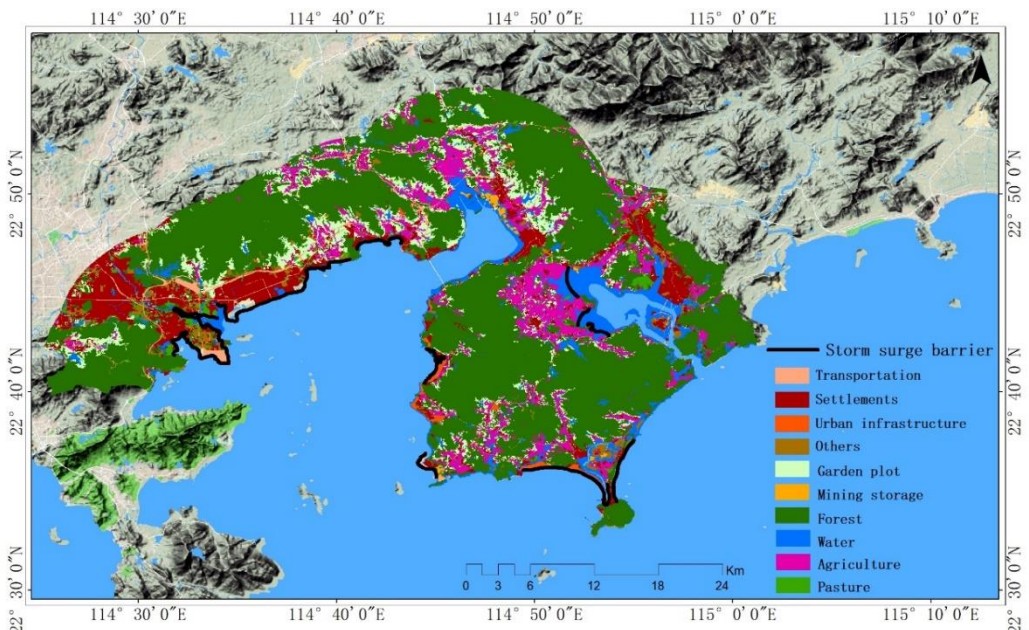

**Figure 15.** The exposure map of different land cover types over the study area. The map was made with ArcGIS 10.5 software based on
the terrain base map layer, which was obtained from Google Maps (Map data ©2019 Google).

Figure 15 shows the distribution of different land cover types over the study area, which provides an overall view of the
region where the forest is concentrated, the zone of settlements, or the location of urban infrastructure.

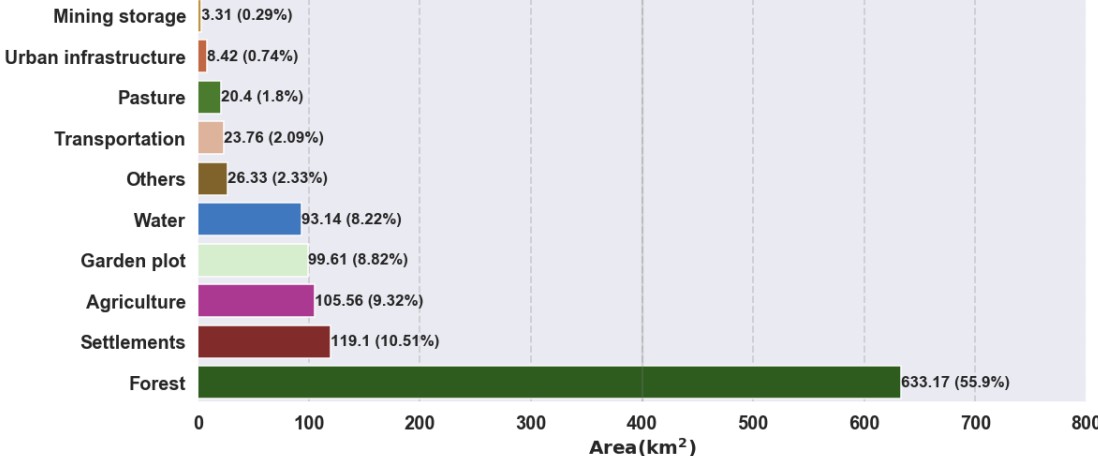

**Figure 16.** The land areas and proportions of different land cover types in the study area.

As seen in Figure 16, the forest land occupies most of the land surface of the study area (55.9%, approximately 633.17
km²). The second-largest land cover type is settlements land, which occupies approximately 10.51% (119.1 km²) of the study





area's surface. The agricultural land, the garden plot land, and the water land have a large surface area, while mining storage
land, the urban infrastructure land, the pasture land, the transportation land, and other lands have a low surface area.

According to the relation between the exposure of land cover types and their corresponding vulnerability values described
in Table 6, the four vulnerability levels with each covered zone to storm surge in the study area can be determined, as
displayed in Figure 17. Moreover, the land area and percentage of each vulnerability zone to the study area were summarized
in Table 8.

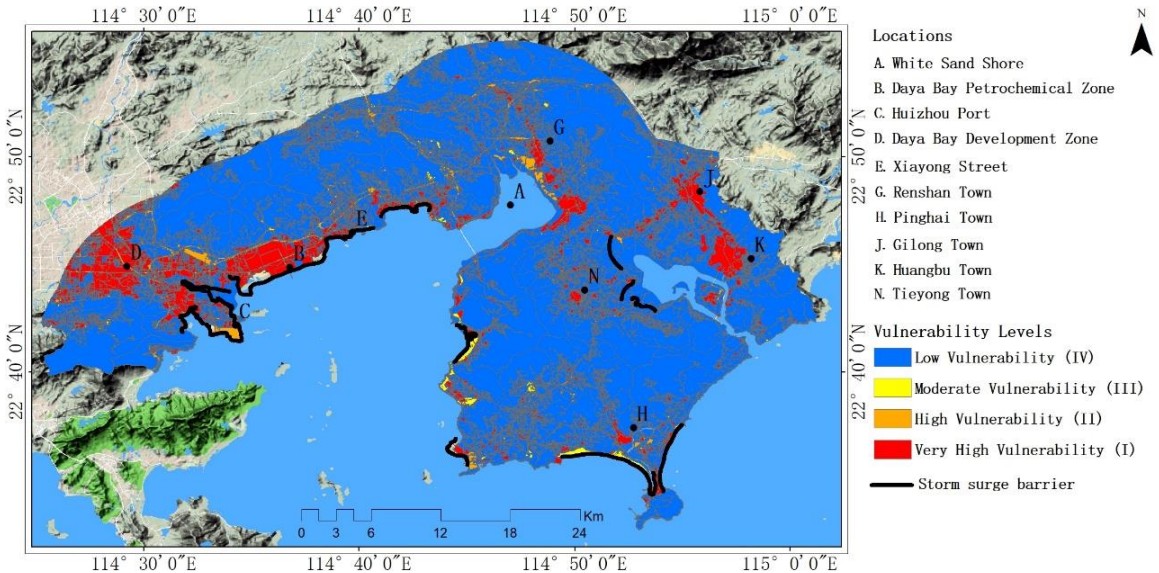


**Figure 17.** The map of vulnerability zones in the study area. The red means the highest vulnerability level (I) and the blue stands for the
lowest vulnerability level (IV) on the map. The map was made with ArcGIS 10.5 software based on the terrain base map layer, which was
obtained from Google Maps (Map data ©2019 Google).

**Table 8.** The vulnerability level, area, and proportion for each vulnerability zone

| Vulerability Level | Color | Area(km²) | Proportion(%) |
|---|---|---|---|
| Very High (I) | Red | 119.10 | 10.51 |
| High (II) | Orange | 27.07 | 2.38 |
| Moderate (III) | Yellow | 8.42 | 0.74 |
| Low (IV) | Blue | 978.21 | 86.37 |

Table 8 shows that the total area is 1132.8 km². The zone marked by low vulnerability level (IV) covers an area of 978.21
km². The zone is mainly present in the forest, agriculture, garden plot, and water, and accounts for the greatest proportion of
the study area. The zone assigned as the highest vulnerability level (I) is covered with settlements and its geographical area
is 119.1 km². The vulnerability level in the land area, which is covered by mining storage and transportation, is high level (II)



and it makes up 2.38% of the study area. The moderate vulnerability (III) zone is mainly under the urban infrastructure class with a total area of 8.42 km$^2$.

Figure 17 is shown that the zones marked by a very high vulnerability (level I) are mainly distributed in the center of the Daya Bay Petrochemical Zone (B) and the Daya Bay Development Zone (D), the residential areas including the Renshan Town (G), the Gilong Town (J), and the Huangbu Town (K). (1). The high-density human settlements and the high concentrated petroleum and chemical industries make the Daya Bay Petrochemical Zone (B) highly susceptible to the effects of surge disaster, and thus the property and human loss caused by the storm surge are significant. (2). The center of the Daya Bay Development Zone (D) could suffer potential human casualties during storm surge events due to the higher density of the population. (3). The residential areas including the Renshan Town (G), the Gilong Town (J), Pinghai Town (H), and the Huangbu Town (K), which lie along the coastline, are more vulnerable to the storm surge without the protection from barriers. Therefore, these zones would most likely suffer significant losses from storm surge.

Furthermore, the Huizhou port (C) is located on the coast, playing a critical role in global trading, which could leave residents and assets with greater exposures to the storm surge than these located on the further inland. Accordingly, the vulnerability level in the Huizhou port (C) is high (level II).

Moreover, some urban infrastructures, transportations, and mining storages that are situated along the coastline of Huizhou are sensitive to the storm surge, and the vulnerability level for these coastal zones is moderate (III) or high (II).

In addition, the most common land cover types over the study area are forest land, agriculture land, the garden plot land, and the water land. These land cover types are hardly affected by natural disasters. Thus, most regions in the study area colored by blue are under the low vulnerability level (IV).

## 4.3 Risk assessment

With the risk matrix approach (Table 7), the risk map in the study area can be made by the combination of the hazard map and the vulnerability map. The risk region was categorized into four dangerous zones represented by different colors, as shown in Figure 18. The statistics of the areas of different risk level zones for each of the design scenarios were summarized in Figure 19.



(a)

(b)

(c)

(d)

(e)

**Figure 18.** The risk assessments maps represent the potential risk of storm surge for five designed storm scenarios ((a):880 hPa/1000-year, (b):910 hPa/100-year, (c):920 hPa/50-year, (d):930 hPa/20-year, (e):940 hPa/10-year). Different colors represent different risk levels. The red means the very high risk, orange stands for high risk, yellow represents moderate risk, and blue indicates low risk on the map. The important locations in the study area are labeled with letters and the violet lines refer to the storm surge barriers along the coastline of Huizhou. These maps were made with ArcGIS 10.5 software based on the terrain base map layer, which was obtained from Google Maps (Map data ©2019 Google).

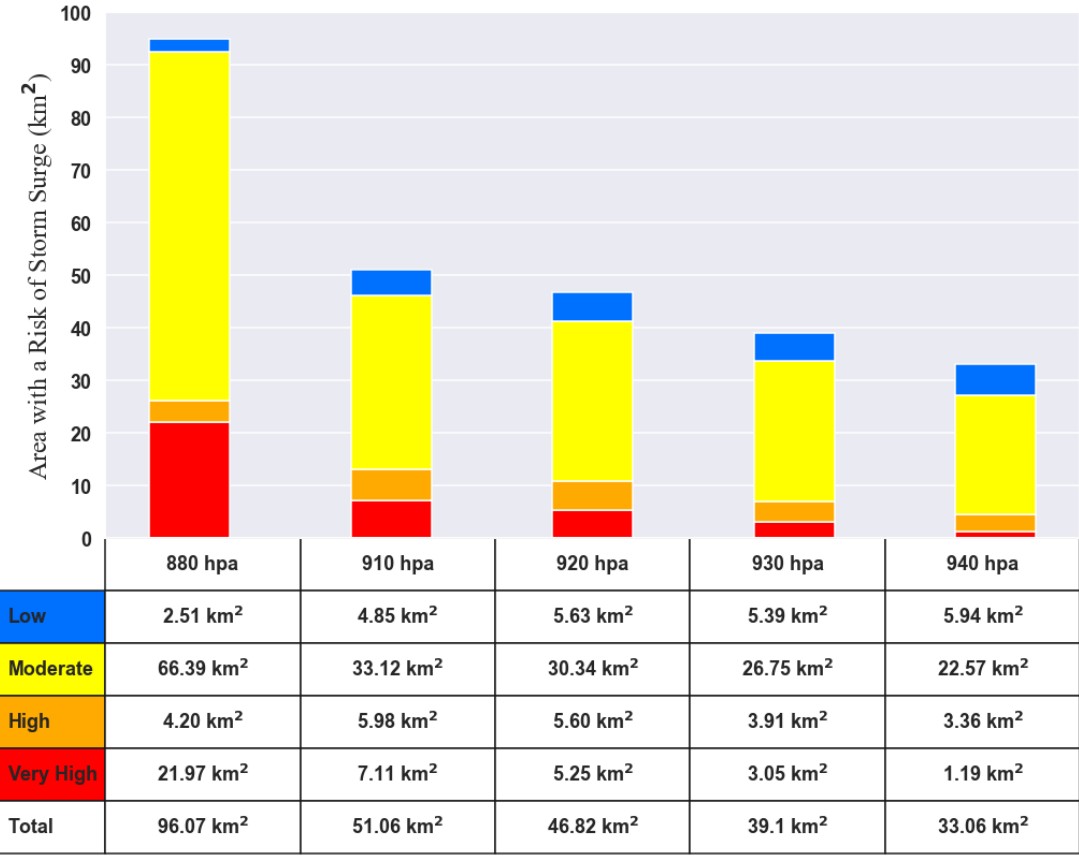

| | 880 hpa | 910 hpa | 920 hpa | 930 hpa | 940 hpa |
|---|---|---|---|---|---|
| Low | 2.51 km² | 4.85 km² | 5.63 km² | 5.39 km² | 5.94 km² |
| Moderate | 66.39 km² | 33.12 km² | 30.34 km² | 26.75 km² | 22.57 km² |
| High | 4.20 km² | 5.98 km² | 5.60 km² | 3.91 km² | 3.36 km² |
| Very High | 21.97 km² | 7.11 km² | 5.25 km² | 3.05 km² | 1.19 km² |
| Total | 96.07 km² | 51.06 km² | 46.82 km² | 39.1 km² | 33.06 km² |

**Figure 19.** The areas of different risk level zones for each of the designed typhoon scenarios (880 hPa, 910 hPa, 920 hPa, 930 hPa, 940 hPa).

The statistics data in Figure 19 indicates that the total area at risk level and the land area labeled with a very high-level decrease with the increasing minimum central pressure of typhoon. The regions under moderate risk level take the largest portion of the total area at risk level for each of typhoon scenarios. For example, approximately 66.39 km² area is exposed to the moderate risk as the minimum central pressure is the 880 hPa, and an area of about 22.57 km² would be at moderate risk when the minimum central pressure is the 940 hPa.

Figure 18 shows that the high concentration of petroleum industries and the high density of the population in the Daya Bay Petrochemical Zone (B) make its vulnerability very high. When the minimum central pressure is 880 hPa, the Daya Bay Petrochemical Zone (B) is in deep inundation status (level I). Thus, the risk level in this zone is at a very high (level I). As the minimum central pressure is 910 hPa or 920 hPa, the risk levels over a wide area of the Daya Bay Petrochemical Zone (B) are high and moderate. However, the Daya Bay Petrochemical Zone (B) is largely free from risk due to the protection provided by the storm surge barriers along the coastline when the minimum central pressure is 930 hPa or 940 hPa.



Moreover, the Huizhou Port (C) and the southeast of the Daya Bay Development Zone (D) are classified as very high or high-risk regions because the regions occupied by transportation land or human settlements (Figure 17) are combined with
high or very high hazard levels to storm surge (Figure 14).

In addition, the regions featuring very high or high-risk levels are the area to the north of the White Sand Shore (A), the Renshan Town (G), Pinghai Town (H), and the Huangbu Town (K). Without the protection from the barrier system, these regions mainly occupied by humans are under moderate risk level even as the minimum central pressure is 940 hPa.

Although the vulnerability level is high in the Gilong Town(J) and the Tieyong Town (N), there is no sign of the risk of
storm surge due to their locations far from the coastline when the minimum central pressure is 910 hPa, 920 hPa, 930 hPa, or 940 hPa.

## 5 Conclusions

In this paper, the application of the coupled model (ADCIRC–SWAN) and the Jelesnianski method for semi-quantifying potential risk assessment of storm surge under different typhoon intensity scenarios in the coastal area of Huizhou. The
typhoon intensity scenarios were designed to the minimum central pressure of 880 hPa, 910 hPa, 920 hPa, 930 hPa, and 940 hPa (corresponding to 1000-year, 100-year, 50-year, 20-year, and 10-year return period). The coastal dikes and levees, which are supposed not damaged during the modelling period, were included in the ADCIRC+SWAN model and Jelesnianski method to simulate the storm surge. The possible inundation extents and depths of storm surge under five different typhoon intensities were computed and the risk assessments were performed based on coastal storm surge maps by using ArcGIS
10.5 software.

The results indicate that the whole Daya Bay Petrochemical Zone and most of the coastal area of Huizhou are not at risk to the storm surge generated by low recurrence interval typhoon (20-year, and 10-year return period) due to the protection provided by coastal dikes and levees. The maximum inundation extents and depths increase with increasing return periods. Significant losses and damages might occur in some parts of the Daya Bay Petrochemical Zone and many coastal
communities for the return periods of 50-year and 100-year scenarios. Moreover, the regions extending from 4 km to 6 km offshore, particularly in the Daya Bay Petrochemical Zone, are under high or very high-risk level to a 1000-year return period typhoon-induced storm surge.

The study provides a comprehensive assessment and zonation of hazard, vulnerability, and risk of storm surge to reduce disaster losses, which caused by designed typhoon scenarios (1000-year, 100-year, 50-year, 20-year, and 10-year return
period) in the coastal area of Huizhou. The risk maps and the risk analysis have been used in practice in Huizhou city, China. The risk maps can help decision-makers in Huizhou recognize the densely populated communities under risk levels and allow them to develop evacuation strategies to minimize civilian casualties. Moreover, the study analyses the storm surge risk especially for the Daya Bay Petrochemical Zone, which is occupied by the high concentration of petroleum industries. The risk analysis can provide a better understanding of the risk regions in the Daya Bay Petrochemical Zone. It can both
reduce economic losses and prevent environmental damage caused by the massive chemical pollutants and oil spills from
coastal petroleum industries that are affected by storm surge. Besides, the proposed methodology and procedure can be
applied to any coastal cities in China for conducting risk assessments of storm surge.

   In further research, the risk assessment should be undertaken in the following aspects:

   (1). The evaluation method based on the different land cover classes is simple. The stage-damage function is regarded as
one of the most effective solutions to storm surge damage assessment. Therefore, the vulnerability curve rather than the land
cover types should be utilized to conduct quantitatively vulnerability assessment in the study area.

   (2). Because of increasing typhoon intensity and rising sea levels caused by climate change in the future, the heightened
storm surge will be taken into consideration when assessing future risk and making hazard mitigation plans in the study area.

   (3). When the maximum inundated depths and extents are calculated under different intensity typhoons, the levee breach
along the coastline will be included in the modelling process to improve simulation precision.

*Data availability*. The typhoon record was obtained from the China Meteorological Administration. The dataset of
Storm Surge Barriers was acquired from the Huizhou Oceanic Administration. The dataset of Water Level Records was
obtained from the Huizhou Oceanic Administration. Other datasets (DEM, Land Cover Types, District Boundaries) were
available from the Huizhou Land and Resources Bureau. The datasets used in the study can be obtained from
https://doi.org/10.6084/m9.figshare.12459794.v1.

*Competing interests*. The authors declare that they have no conflict of interest.

*Author contributions*. Lin Mu designed the concept; Zhenfeng Yao and Si Wang collected the data in this study; Si Wang
and Jia Gao completed the coding and carried out experiments; Lin Mu and Si Wang analyzed experimental results; Si Wang,
Lin Mu, and Enjing Zhao wrote the manuscript.

*Financial support*. This work was supported by the Disaster Prevention and Reduction Project for Huizhou Fishery and
Aquaculture Industry (Grant No. 2018HZBXSZ04027), the National Key Research and Development Program of China
(Grant No. 2017YFC1404700) and the Discipline Layout Project for Basic Research of Shenzhen Science and Technology
Innovation Committee (Grant No. JCYJ20170810103011913).

*Acknowledgements*. Thanks to the South China Sea Marine Prediction Center whose researchers helped to evaluate the
performance of the coupled model (ADCIRC-SWAN) by comparing the predicted water levels and the observed water levels
in the study area and simulate the possible inundation depths and extents of typhoon surge with the coupled model for the
designed typhoon scenarios.



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
