# Peer review of "Assessing and zoning of typhoon storm surge risk with GIS technique"

_Natural Hazards and Earth System Sciences, 2020_

## Referee Comment (RC1) · Anonymous Referee #1 · 28 Jun 2020

The study conducted the assessment and zonation of typhoon storm surge hazard, vulnerability, and risk in a case study area with a high concentration of petroleum industries and population density, finding regions with different risk levels in the study area under five representative typhoon scenarios. These risk maps and analysis can aid in developing storm surge management strategies and evacuation plans, and the methodology (hydrological and wave models; GIS technique; exposure analysis) and the procedure can be applied in other locations. However, I found some issues about methodology and figure presentation that should be addressed in the discussion section to improve the quality of the article, which are summarized below.

Major concerns: (1) In the study, the performance of the coupled model for storm surge modeling has already been validated by comparing the simulated data and the water level records obtained from the Huizhou gauging station. However, the performance assessment of the coupled model was not done properly because the Huizhou gauging station was on the left of the study area. I encourage the authors to make another comparison between simulated data and recorded water levels that were obtained from one gauging station on the right of the study area. The validated results from these two gauging stations can make the performance of the coupled model more convincing over the study area.

(2) The storm surge modeling with the coupled model is an important step for this research and the modeling section requires more clarifying. For the ADCIRC model, I suggest authors give more description including what the discrete method was used and which coordinate system was chosen in the ADCIRC model, and how do you consider the bottom friction in the ADCIRC model because the different land types have various frictional values during the storm surge modeling.

(3) The data analysis of storm surge risks on different towns in the study area and the relation between hazard assessment and risk assessment can be added to the research paper, which is helpful to readers, especially to decision-makers, to better understand the impact from the storm surge.

(4). The authors conducted the risk assessment of storm surge for Huizhou city and we can identify the risk zones on different intensity scenarios from the results (analysis and maps). I encourage the authors to demonstrate how the results are useful for developing risk response plans and evacuation strategies for storm surge risk in Huizhou. It is a crucial aspect that can strengthen the study and the manuscript.

(5) I suggest the authors improve the presentation of figures 14, 15, 17, and 18: The terrain base map layer can be transparent to avoid blending with the colors on the assessment map layer. The font size of texts on the assessment map layers should

be increased. The legends in figures can be removed because they are repeated many times in figures 14 and 18 (b), (c), (d), and (e). The data analysis figure about vulnerability assessment can be added after figure 17 in the paper. The administrative boundaries might be displayed in the figures.

Minor comments on specific lines: 2.1 Study area 1) Figure 1: It is not clear to me where the Daya Bay Petrochemical Zone is located. It is important to add a visualization of the Daya Bay Petrochemical Zone in Figure 1 because the Petrochemical Zone with a high concentration of petroleum facilities is the reason to conduct the hazard and risk assessment of storm surge in the study.

2) L130- L145: Some information about the study area is not necessary and can be removed.

3) L160: The elevation map of the study area which is a crucial factor for the storm surge modeling in the paper should be displayed.

4) Figure 2: The more visualization about the barrier that is an important aspect might be given.

5) L178: Administrative boundaries at the township level should be displayed in the figure.

6) Figure 2 and Figure 3: North arrow in these Figures are not clear.

3.1 Model description and validation 1) L274: The unit of the radius of maximum wind (Rmax) on the Y-axis should be 'km' in Figure 7.

4 Results and discussion 1) Figure 13: The legends are not clear.

2) Figure 16: The colors of different bars can be set to the same value

---

## Referee Comment (RC2) · Anonymous Referee #2 · 29 Jun 2020

The study by Wang et al. provides a case study to assess typhoon storm surge risk with GIS technique for Huizhou, China. This study is conducted referring to the latest China national rules: Standard Technical Guideline for risk assessment and zoning of marine disaster (Part 1: 80 storm surge) (Ministry of Natural Resources of the People's Republic of China, 2019), by utilizing the Jelesnianski method, coupled ADCIRC-SWAN model and ArcGIS software. Wang et al. have made assessing and zoning of typhoon storm surge risk on Huizhou, with respect to hazard, vulnerability and risk, quantitively.

In the manuscript, the study background of typhoon storm surge risk has been well introduced, as well as proper description of the results. In comparison, the methodology

part needs to be improved by adding useful details with the corresponding references in necessary. Moreover, since this work aims to help decision-makers in Huizhou, it would be useful to see constructive suggestions by the authors, to have this work also as a case for knowledge-to-policy transition. This is also a scope of NHESS. In addition, linguistic improvement is suggested.

Major comments: (1) Line 84: The 'Standard Technical Guideline' is made on Year 2019, how does it use for Year 2016? If such rules have been tested or used before the publication of Guideline, please reconstruct this sentence for proper statement.

(2) In Figure 1, to add a mark for 'Daya Bay'.

(3) Line 130-145 presents the status of Huizhou City, please provide reference or source link.

(4) In 2.2 Dataset requirement, please provide reference for each dataset.

(5) Would it be possible to merge Figure 2 and 3 to one figure?

(6) Line 195, please provide references for these existing studies.

(7) Line 204-205, could you provide further details about the 11 astronomical tidal components? And only tidal components?

(8) Figure 4, the red rectangular in Panel a is different from the region of Panel b and c.

(9) In Lines 238-240: 'The Absolute Error (AE) is computed when the highest measured water level is above 100 cm. The Relative Error (RE) is calculated as the measured observed water level is below 100 cm.' Is it consistent with that in Table 2?

(10) Please provide references to Equations in 3.2.2

(11) What meaning are the colors of the curves in Figure 9?

(12) What the meaning of the green and red dots in Figure 10?

(13) The 'Value' in Table 6 is a try of set up via this work, or refers to some references?

Minor comments: (1) Line 6: to use 'concentrations'.

(2) Line 39: should it be the word 'severe'?

(3) Line 44: Please use 'RMB' instead of 'Yuan' throughout the manuscript.

(4) Line 48: Please use a proper reference for IPCC report.

(5) Line 63: impact 'on', not 'to'.

(6) Line 79: Is there a reference for the 'Standard Technical Guideline'? or webpage?

(7) Line 101: Use 'depicts' instead of 'details'.

(8) Line 237: Use 'are' instead of 'were'.

(9) In Table 5 column 1st, the bracket is not paired in format.

(10) Line 527: replace 'increasing' by 'lowering'.

(11) Line 577: Use increased instead of heightened.

---

## Author Comment (AC1) · 8 Jul 2020

We would like to thank Referee #2 for reviewing our manuscript and giving constructive comments and suggestions, which substantially helped us improve the quality of the paper. Appropriated changes have been introduced to the revised manuscript according to the referee's comments and suggestions. In the following, responses to the referee's comment are described in a point-to-point manner. The referee's comments are displayed in the grey background and the responses are displayed in blue.

**Major comments:**

(1) Line 84: The 'Standard Technical Guideline' is made on Year 2019, how does it use for Year 2016? If such rules have been tested or used before the publication of Guideline, please reconstruct this sentence for proper statement.

**Response:** The trial version of the *technical directives for risk assessment and zoning of marine disaster-Part 1: storm surge* was published in 2012. In this paper, the risk assessment of storm surge was conducted with the latest version (2019). This sentence has been reconstructed in the revised manuscript.

(2) In Figure 1, to add a mark for 'Daya Bay'.

**Response:** Figure 1 in the original manuscript has been removed, and the remade Figure 1 in which the 'Daya Bay Petrochemical Zone' and the 'petrochemical building distribution in the Zone' were added is as follows. Moreover, according to the comments about the figure presentation by the two referees, many figures have been remade at high resolution, as can be seen in the revised manuscript.

[Figure]

(3) Line 130-145 presents the status of Huizhou City, please provide reference or source link.

**Response:** The source links (Huizhou Guangdong province, 2019; Huizhou, 2018) which provide a detailed introduction of Huizhou City and study area in Line 130-145 have been added in the revised manuscript.

(4) In 2.2 Dataset requirement, please provide reference for each dataset.

**Response:** The reference for the dataset in section 2.2 has been added in the revised manuscript and the datasets can be downloaded from the link (nhess-datasets-2020-130, 2020).

(5) Would it be possible to merge Figure 2 and 3 to one figure?

**Response:** Figure 2 and Figure 3 in the original manuscript were merged into the Figure as follows. The different sections of storm surge barriers along the coastline of the study area and two gauging stations including Huizhou gauging station and Gangkou gauging station were marked in the Figure as shown below. The lines and dots represent the storm surge barriers and gauging stations, respectively in the Figure. The remade Figure has been added in the revised manuscript as Figure 3.

[Figure]

(6) Line 195, please provide references for these existing studies.

**Response:** The references (Kerr et al., 2013; Orlić et al., 2010; Li et al., 2020) about the hydrology model applied in the regions have been added in the revised manuscript.

(7) Line 204-205, could you provide further details about the 11 astronomical tidal components? And only tidal components?

**Response:** The open boundary of the model water level is controlled by the total water level, which is obtained by the superposition of 11 astronomical tidal components. These 11 astronomical tidal components are M2, N2, S2, K2, K1, O1, P1, Q1, $MS_4$, $M_4$, $M_6$.

(8) Figure 4, the red rectangular in Panel a is different from the region of Panel b and c.

**Response:** The red rectangular spanning from 113˚W to 116˚W longitude in the original Figure 4 was replaced with the rectangular spanning from 114˚W to 116˚W longitude in the

revised manuscript, which is exactly corresponded to the region in the Panel b and c. The remade Figure is as follows.

[Figure]

[Figure]

(9) In Lines 238-240: 'The Absolute Error (AE) is computed when the highest measured water level is above 100 cm. The Relative Error (RE) is calculated as the measured observed water level is below 100 cm.' Is it consistent with that in Table 2?

**Response:** Thanks for your correction. The sentence was written wrong in Lines 238-240. The rewritten text was added in the revised manuscript that the Absolute Error (AE) is computed when the highest measured water level is **below** 100 cm and the Relative Error (RE) is calculated as the measured observed water level is above 100 cm, which is consistent with statistics in Table 2

(10) Please provide references to Equations in 3.2.2

**Response:** The references (Vickery et al., 2000; Cheung et al., 2007) on the Equations in 3.2.2 have been added in the revised manuscript.

(11) What meaning are the colors of the curves in Figure 9?

**Response:** Abbreviations that Tropical Depression (TD), Tropical Storm (TS), Severe Tropical Storm (STS), Typhoon (TY), Severe Typhoon Super (STY), and Super Typhoon (Super TY) are explained in detail in the Figure 9 caption. The rewritten Figure 9 caption has been added in the revised manuscript.

(12) What the meaning of the green and red dots in Figure 10?

**Response:** The remade Figure containing figure legend in which the different colors represent wind speeds has been added in the revised manuscript. The remade Figure is as follows.

[Figure]

(13) The 'Value' in Table 6 is a try of set up via this work, or refers to some references?

**Response:** The vulnerability values of land types to storm surge are provided by the guideline, which can be downloaded from the link (Ministry of Natural Resources of the People's Republic of China, 2019).

**Minor comments:**

(1) Line 6: to use 'concentrations'.

**Response:** Thanks for your correction.

(2) Line 39: should it be the word 'severe'?

**Response:** Thanks for your correction. The 'Server' has been rewritten with 'severe'.

(3) Line 44: Please use 'RMB' instead of 'Yuan' throughout the manuscript.

**Response:** Thanks for your correction. The 'Yuan' has been replaced with 'RMB'.

(4) Line 48: Please use a proper reference for IPCC report.

**Response:** Thanks for your suggestion.

(5) Line 63: impact 'on', not 'to'.

**Response:** Thanks for your correction. The 'to' has been replaced with 'on'.

(6) Line 79: Is there a reference for the 'Standard Technical Guideline'? or webpage?

**Response:** Thanks for your correction. The reference was added in the revised manuscript.

(7) Line 101: Use 'depicts' instead of 'details'.

**Response:** Thanks for your correction. The 'details' has been replaced with 'depicts'.

(8) Line 237: Use 'are' instead of 'were'.

**Response:** Thanks for your correction. The 'were' has been replaced with 'are'.

(9) In Table 5 column 1st, the bracket is not paired in format.

**Response:** Thanks for your correction.

(10) Line 527: replace 'increasing' by 'lowering'.

**Response:** Thanks for your correction. The 'increasing' has been replaced with 'lowering'

(11) Line 577: Use increased instead of heightened.

**Response:** Thanks for your correction. The 'heightened' has been replaced with 'increased'

**References**

(1) Huizhou Guangdong province, 2019, available online at: http://govt.chinadaily.com.cn/s/201907/08/WS5d145d96498e5314096b63f5/huizhou-guangdong-province.html.

(2) Huizhou, 2018, available online at: https://www.bayarea.gov.hk/en/about/huizhou.html.

(3) nhess-datasets-2020-130, 2020, available online at: https://doi.org/10.6084/m9.figshare.12459794.v1.

(4) Kerr, P. C., Donahue, A. S., Westerink, J. J., Luettich Jr, R. A., Zheng, L. Y., Weisberg, R. H., ... & Roland, A. (2013). US IOOS coastal and ocean modeling testbed: Inter-model evaluation of tides, waves, and hurricane surge in the Gulf of Mexico. Journal of Geophysical Research: Oceans, 118(10), 5129-5172.

(5) Orlić, M., Belušić, D., Janeković, I., & Pasarić, M. (2010). Fresh evidence relating the great Adriatic surge of 21 June 1978 to mesoscale atmospheric forcing. Journal of Geophysical Research: Oceans, 115(C6).

(6) Li, A., Guan, S., Mo, D., Hou, Y., Hong, X., & Liu, Z. (2020). Modeling wave effects on storm surge from different typhoon intensities and sizes in the South China Sea. Estuarine, Coastal and Shelf Science, 235, 106551.

(7) Vickery, P. J., Skerlj, P. F., & Twisdale, L. A. (2000). Simulation of hurricane risk in the US using empirical track model. Journal of structural engineering, 126(10), 1222-1237.

(8) Cheung, K. F., Tang, L., Donnelly, J. P., Scileppi, E. M., Liu, K. B., Mao, X. Z., ... & Murnane, R. J. (2007). Numerical modeling and field evidence of coastal overwash in southern New England from Hurricane Bob and implications for paleotempestology. Journal of Geophysical Research: Earth Surface, 112(F3).

(9) Ministry of Natural Resources of the People's Republic of China, Technical guidance for risk assessment and zonation of storm surge disaster, 2019, available online at: https://www.re nrendoc.com/p-82139795.html  (in Chinese).

---

## Referee Comment (RC3) · Anonymous Referee #2 · 11 Jul 2020

Wang and co-authors have addressed all my comments from my last-round review into the revised manuscript. This paper provides a study for the up-to-date assessing and zoning of typhoon storm surge risk with GIS technique in China, fitting the scope of the journal.

No further comments from my side. Here, I suggest this manuscript to be published in NHESS.

---

## Author Comment (AC2) · 13 Jul 2020

We thank Referee #1 for the careful reading and reviewing our manuscript entitled: "Assessing and zoning of typhoon storm surge risk with GIS technique: A case study of the coastal area of Huizhou". We sincerely appreciate these constructive comments and valuable suggestions for improving our manuscript. In the following, responses to the referee's comment are described in a point-to-point manner. The referee comments are displayed in the grey background and the responses are displayed in blue.

**Major comments:**

(1) In the study, the performance of the coupled model for storm surge modeling has already been validated by comparing the simulated data and the water level records obtained from the Huizhou gauging station. However, the performance assessment of the coupled model was not done properly because the Huizhou gauging station was on the left of the study area. I encourage the authors to make another comparison between simulated data and recorded water levels that were obtained from one gauging station on the right of the study area. The validated results from these two gauging stations can make the performance of the coupled model more convincing over the study area.

**Response:** Figure 2 and Figure 3 in the original manuscript were merged into the Figure. The Gangkou gauging station on the eastern side of the study area has been added to the map, as shown below.

Figure. The two stations including Huizhou gauging station and Gangkou gauging station measure the water levels in the study area.

Moreover, the measured water levels from the two gauging stations including Huizhou gauging station and Gangkou gauging station were compared with the simulated water levels during ten storm

**events, as shown below.**

---

## Author Comment (AC3) · 4 Sep 2020

We would like to thank the reviewers for the thoughtful review of our manuscript and for giving these constructive comments and suggestions, which substantially helped us improve the quality of the paper. Almost all the points that were raised have been adopted in the revised manuscript. We believe the new version of the manuscript has been significantly improved.

---

## Author Response (AR1)

Dear Editors and Reviewers,

Thank you very much for the review of our manuscript entitled: "Assessing and zoning of typhoon storm surge risk with GIS technique: A case study of the coastal area of Huizhou".

We sincerely appreciate all the comments and suggestions made by the editors and reviewers. These comments and suggestions are all constructive and valuable, which helped us to improve the quality of the article. We have carefully discussed the comments, performed additional experiments, analyzed these experiments, and made corrections according to comments from the editors and the reviewers. The appropriated changes suggested by the reviewers have been introduced to the revised manuscript.

The introduced major changes with section, page, and line in marked-up manuscript are listed below. Minor changes are marked in the marked-up manuscript. We hope that these revisions are clear enough to follow.

**The list of changes:**

1). Co-author and affiliation. Professor Wang provided some detailed skills and approaches in the experiment to improve the quality of our manuscript during the revision stage. Thus, we added him as a co-author in the revised manuscript.

2). Abstract. (page 3, line 82). The sentence that a panel of Chinese ocean disaster prevention and reduction researchers updated and published the latest standard Technical guideline for risk assessment and zoning of marine disaster (Part 1: storm surge) has been reconstructed in the revised manuscript, as pointed out by Reviewer 2.

3). Section 2.1. (page 6). 'Daya Bay' has been marked in Figure 1, as pointed out by Reviewer 1 and Reviewer 2.

4). Section 2.1. (page 7, lines 145-146). The source links related to Huizhou City have been added in the revised manuscript, as recommended by Reviewer 2.

5). Section 2.2. (page 7, line 156). The reference for the dataset has been added to the revised manuscript, as recommended by Reviewer 2.

6). Section 2.2. (page 8). The DEM map of the study area including rivers and the elevation value that point covers, which is shown as Figure 3, has been added to the revised manuscript, as recommended by Reviewer 1.

7). Section 2.2. (page 10). Figure 2 and Figure 3 in the previous manuscript have been merged into Figure 3 in the revised manuscript, as recommended by Reviewer 1 and Reviewer 2.

8). Section 2.2. (page 10, lines 185-200). The information related to the Gangkou gauging station on the eastern side of the study area has been added to the revised manuscript.

9). Section 3.1. (page 11, line 217). The references about the hydrology model applied to the regions have been added to the revised manuscript, as pointed out by Reviewer 2.

10). Section 3.1. (pages 11-12, lines 217-224). The bottom friction in the ADCIRC model has been added to the revised manuscript, as pointed out by Reviewer 2.

11). Section 3.1. (page 12, line 235). The details about the 11 astronomical tidal components have been added to the revised manuscript, as pointed out by Reviewer 2.

12). Section 3.1. (page 13). Figure 5 (a) has been remade, red rectangular spanning from 113˚W to 116˚W longitude has been replaced with the rectangular spanning from 114˚W to 116˚W longitude in the revised manuscript.

13). Section 3.1. (pages 16-20). The Relative Error (RE) and the Absolute Error (AE) were calculated between observed data and the simulated data obtained from the Gangkou gauging station on the eastern side of the study area, as shown in Table 3. The result can make the performance of the coupled model more convincing over the study area, which has been added to the revised manuscript, as recommended by Reviewer 1.

14). Section 3.1. (page 19, lines 285-287). The sentence that the Absolute Error (AE) is computed when the highest measured water level is below 100 cm and the Relative Error (RE) is calculated as the measured observed water level is above 100 cm has been rewritten, which has been added to the revised manuscript, as pointed out by Reviewer 2.

15). Section 3.2.2. (page 22). The unit in Figure 7 has been corrected, as pointed out by Reviewer 1.

16). Section 3.2.4. (page 25). The abbreviations are explained in detail in Figure 9 caption, as pointed out by Reviewer 2.

17). Section 3.2.4. (page 26). Figure 10 has been remade, which contains legend to convey the meaning of dots, as pointed out by Reviewer 2.

18). Section 4.1. (page 36). Figure 14 has been remade. The transparency of the terrain base map layer has been set to 50% to avoid blending with colors in the other layers, which makes the figure clear, as pointed out by Reviewer 2.

19). Section 4.1. (page 37, lines 502-516). Figure 15, which shows the simulated inundated areas with different hazard levels for each of the scenarios, has been made and added to the revised manuscript, as pointed out by Reviewer 1.

20). Section 4.2. (pages 38-41). Figure 16, Figure 17, and Figure 18 have been made. The legend in Figure 16 has been moved and the bars in Figure 17 are labeled with different colors according to the vulnerability levels of land cover types, as recommended by the Reviewer 1.

21). Section 4.3. (pages 43-47). Figure 19 and Figure 20 have been remade and Figure 21 has been made, as recommended by the Reviewer 1.

22). Section 4.3. (page 47, lines 631-637). The statistical analysis of the relationship between hazard assessment and risk assessment over the study area has been added to the revised manuscript.

**Response to the Reviewer 1:**

We thank Reviewer 1 for the careful reading and reviewing our manuscript entitled: "Assessing and zoning of typhoon storm surge risk with GIS technique: A case study of the coastal area of Huizhou". We sincerely appreciate these constructive comments and valuable suggestions for improving our manuscript. In the following, responses to the reviewer's comment are described in a point-to-point manner. The reviewer's comments are displayed in the grey background and the responses are displayed in blue.

**Major comments:**

(1) In the study, the performance of the coupled model for storm surge modeling has already been validated by comparing the simulated data and the water level records obtained from the Huizhou gauging station. However, the performance assessment of the coupled model was not done properly because the Huizhou gauging station was on the left of the study area. I encourage the authors to make another comparison between simulated data and recorded water levels that were obtained from one gauging station on the right of the study area. The validated results from these two gauging stations can make the performance of the coupled model more convincing over the study area.

**Response:** Thanks for your valuable suggestion. Figure 2 and Figure 3 in the original manuscript were merged into the new Figure 3 in the revised manuscript. The Gangkou gauging station on the eastern side of the study area has been added to the map, as shown in Figure 1.

[Figure]

**Figure 1**. The two stations including Huizhou gauging station and Gangkou gauging station measure the water levels in the study area.

Moreover, the measured water levels from the two gauging stations including Huizhou gauging station and Gangkou gauging station were compared with the simulated water levels during ten storm events have been added to the revised manuscript, as shown in Figure 2.

[Figure]

[Figure]

**Figure 2**. The predicted water levels (in black line) and highest measured water levels (in red dots) recorded by the Huizhou station and Gangkou station during the typhoon events.

In addition, the Relative Error (RE) is calculated as the measured observed water level is above 100 cm. The Absolute Error (AE) is computed when the highest measured water level is below 100 cm. The statistical results from two stations have been summarized in Table 1. It displays that the data with RE≤20%

or AE≤20 cm account for 90% of all simulated data, which satisfies the criterion in the guideline.

**Table 1.** Summary of error statistics (AE and RE) between maximum predicted water levels and highest measured water levels from Huizhou station and Gangkou station during the typhoon events.

| Typhoon Name | Measured Data (cm) | | Relative Error (%) | | Absolute Error (cm) | |
|---|---|---|---|---|---|---|
| | Huizhou | Gangkou | Huizhou | Gangkou | Huizhou | Gangkou |
| Nuri (0812) | 129 | 84 | 2 | / | / | 15 |
| Hagupit (0814) | 135 | 126 | 6 | 2 | / | / |
| Molave (0906) | 116 | 58 | 13 | / | / | 17 |
| Vicente (1208) | 136 | 87 | 18 | / | / | 11 |
| Usagi (1319) | 137 | 102 | 21 | 17 | / | / |
| Nida (1604) | 92 | 94 | / | / | 18 | 2 |
| Haima (1622) | 81 | 94 | / | / | 18 | 12 |
| Hato (1713) | 120 | 81 | 30 | / | / | 0 |
| Khanun (1720) | 98 | 93 | / | / | 5 | 14 |
| Mangkhut (1822) | 278 | 151 | 10 | 1 | / | / |

(2) The storm surge modeling with the coupled model is an important step for this research and the modeling section requires more clarifying. For the ADCIRC model, I suggest authors give more description including what the discrete method was used and which coordinate system was chosen in the ADCIRC model, and how do you consider the bottom friction in the ADCIRC model because the different land types have various frictional values during the storm surge modeling.

**Response:** Thanks for your suggestion. We highly appreciate the reviewer's comment. The ADCIRC model uses the finite element method allowing the utilizing of flexible and unstructured grids. The ADCIRC model can be applied in modeling tides, wind-driven circulation, and typhoon-generated storm surge. The ADCIRC-2D in this study was run using a spherical coordinate system. It can provide both water surface elevation and the depth-averaged velocity fields with employing continuity equation and momentum equations.

For the friction coefficient, the ADCIRC model in this study utilized Manning's n values derived from the dataset of land cover types. The dataset contains information about the types of land-cover associating with Manning's values over the study area. The land types over the study area were associated with Manning's values, and then average the Manning's n values for the ADCIRC mesh, as shown in Table 2.

**Table 2.** The land-cover associating with Manning's values

| Land types | Settlements | Forest | Dryland | Paddy field | Road | Riverway | Open spaces |
|---|---|---|---|---|---|---|---|
| Manning's values | 0.07 | 0.065 | 0.06 | 0.05 | 0.035 | 0.025-0.035 | 0.035 |

(3) The data analysis of storm surge risks on different towns in the study area and the relation between hazard assessment and risk assessment can be added to the research paper, which is helpful to readers, especially to decision-makers, to better understand the impact from the storm surge.

**Response:** Thanks for your suggestion. The total area in hazard and that in risk under different typhoon intensities were shown in Figure 3. It indicates that the area in high hazard level cannot represent that in high-risk level and the area in high hazard level could be turned into the area in moderate or low-risk level due to different vulnerabilities of land type covers. The Figure 3 has been added to the revised manuscript

[Figure]

**Figure 3.** The total area of hazard and risk of storm surge for each of the designed typhoon scenarios (880 hPa, 910 hPa, 920 hPa, 930 hPa, and 940 hPa).

(4). The authors conducted the risk assessment of storm surge for Huizhou city and we can identify the risk zones on different intensity scenarios from the results (analysis and maps). I strongly encourage the authors to demonstrate how the results are useful for developing risk response plans and evacuation strategies for storm surge risk in Huizhou. It is a crucial aspect that can strengthen the study and the manuscript.

**Response:** Thanks for your suggestion. The risk assessment, risk maps, and risk analysis for the study area were made in 2019 and have been used in practice. With these escape routes on the risk maps and risk suggestions, the decision-makers can prioritize development strategies and make evacuation plans in the risk areas of Huizhou city. However, limited by paper length, these dozens of storm surge escape route maps for the study area were provided from the source link (https://figshare.com/articles/figure/escape_routine/12647105).

(5) I suggest the authors improve the presentation of figures 14, 15, 17, and 18: The terrain base map layer can be transparent to avoid blending with the colors on the assessment map layer. The font size of texts on the assessment map layers should be increased. The legends in figures can be removed because they are repeated many times in figures 14 and 18 (b), (c), (d), and (e). The data analysis figure about vulnerability assessment can be added after figure 17 in the paper. The administrative boundaries might be displayed in the figures.

**Response:** Thanks for your suggestion. According to the comments about the presented figures, many figures have been remade, which can be seen in the revised manuscript. The Figures 14, 15, 17, and 18 in the original manuscript were remade and figure depicts the total areas of inundated areas with different hazard levels has been made, as shown in Figure 4. Based on the comments from the referees, the transparency of the terrain base map layer has been set to 50% to avoid blending with colors in the other layers. Moreover, the font size of texts has been increased and the repeated legends were removed. In addition, administrative boundaries have been added to figures.

[Figure]

**Figure 4.** The total areas of inundated areas with different hazard levels under five storm scenarios.

[Figure]

**Figure 5.** The hazard assessment maps represent the inundation extents and depths for five storm scenarios ((a):880 hPa/1000-year, (b):910 hPa/100-year, (c):920 hPa/50-year, (d):930 hPa/20-year, (e):940 hPa/10-year).

[Figure]

**Figure 6.** The exposure map of different land cover types over the study area.

[Figure]

**Figure 7.** The map of vulnerability zones in the study area. The red means the highest vulnerability level (I) and the blue stands for the lowest vulnerability level (IV) on the map.

[Figure]

**Figure 8.** The risk assessments maps represent the potential risk of storm surge for five designed storm scenarios

**Minor comments:**

2.1 Study area

1) Figure 1: It is not clear to me where the Daya Bay Petrochemical Zone is located. It is important to add a visualization of the Daya Bay Petrochemical Zone in Figure 1 because the Petrochemical Zone with a high concentration of petroleum facilities is the reason to conduct the hazard and risk assessment of storm surge in the study.

**Response:** Thanks for your correction. Figure 1 in the original manuscript has been removed, and the remade Figure 1 in which the 'Daya Bay Petrochemical Zone' and the 'petrochemical building distribution in the Zone' were added to the revised manuscript.

[Figure]

**Figure 9.** The maps of observation locations used in the study: (a) The map of Guangdong Province in the southern area of China (b) The map of Huizhou city and the study area; (c) The map of towns and their boundaries in the study area and the petrochemical buildings are distributed in the Data Bay Petrochemical Zone.

2) L130- L145: Some information about the study area is not necessary and can be removed.

**Response:** Thanks for your suggestion.

3) L160: The elevation map of the study area which is a crucial factor for the storm surge modeling in the paper should be displayed.

**Response:** Thanks for your correction. The DEM map of the study area including rivers and the elevation value that point covers, which is shown as Figure 10, has been added to the revised manuscript

[Figure]

**Figure 10.** The DEM map of the study area including rivers and the elevation value that point covers.

4) Figure 2: The more visualization about the barrier that is an important aspect might be given.

**Response:** Thanks for your suggestion. These storm surge barriers have been added to the figure in the revised manuscript.

5) L178: Administrative boundaries at the township level should be displayed in the figure.

**Response:** Thanks for your correction. Administrative boundaries at the township level have been added to the figures in the revised manuscript.

6) Figure 2 and Figure 3: North arrow in these Figures are not clear.

**Response:** Thanks for your correction. The North arrow in Figure 2 and Figure 3 have been remade.

3.1 Model description and validation

1) L274: The unit of the radius of maximum wind (Rmax) on the Y-axis should be 'm' in Figure 7.

**Response:** Thanks for your correction. The unit in Figure 7 has been replaced with 'm' in the revised manuscript.

[Figure]

**Figure 11.** The relation between the pressure difference ($\Delta P$) and the radius of maximum wind ($R_{max}$).

4 Results and discussion

1) Figure 13: The legends are not clear.

**Response:** Thanks for your correction. The legends have been remade in the revised manuscript.

2) Figure 16: The colors of different bars can be set to the same value

**Response:** Thanks for your correction. Figure 16 has been remade. The bars are labeled with different colors according to the vulnerability levels of land cover types, as shown in Figure 12.

[Figure]

**Figure 12**. The land areas and proportions of different land cover types in the study area.

**Response to the Reviewer 2:**

We would like to thank Reviewer 2 for reviewing our manuscript and giving constructive comments and suggestions, which substantially helped us improve the quality of the paper. Appropriated changes have been introduced to the revised manuscript according to the reviewer's comments and suggestions. In the following, responses to the reviewer's comment are described in a point-to-point manner. The reviewer's comments are displayed in the grey background and the responses are displayed in blue.

**Major comments:**

(1) Line 84: The 'Standard Technical Guideline' is made on Year 2019, how does it use for Year 2016? If such rules have been tested or used before the publication of Guideline, please reconstruct this sentence for proper statement.

**Response:** Thanks for your valuable suggestion. The trial version of the *technical directives for risk assessment and zoning of marine disaster-Part 1: storm surge* was published in 2012. In this paper, the risk assessment of storm surge was conducted with the latest version (2019). This sentence has been reconstructed in the revised manuscript.

(2) In Figure 1, to add a mark for 'Daya Bay'.

**Response:** Thanks for your valuable suggestion. Figure 1 in the original manuscript has been removed. The remade figure, as shown in Figure 1, in which the 'Daya Bay Petrochemical Zone' and the 'petrochemical building distribution in the Zone' were marked, has been added to the revised manuscript. as can be seen in the revised manuscript.

[Figure]

**Figure 1.** The maps of observation locations used in the study: (a) The map of Guangdong Province in the southern area of China (b) The map of Huizhou city and the study area; (c) The map of towns and their boundaries in the study area and the petrochemical buildings are distributed in the Data Bay Petrochemical Zone.

(3) Line 130-145 presents the status of Huizhou City, please provide reference or source link.

**Response:** Thanks for your valuable suggestion. The source links (Huizhou Guangdong province, 2019; Huizhou, 2018) which provide a detailed introduction of Huizhou City and study area in Line 130-145 have been added in the revised manuscript.

(4) In 2.2 Dataset requirement, please provide reference for each dataset.

**Response:** Thanks for your valuable suggestion. The reference for the dataset in section 2.2 has been added in the revised manuscript and the datasets can be downloaded from the link (nhess-datasets-2020-130, 2020).

(5) Would it be possible to merge Figure 2 and 3 to one figure?

**Response:** Thanks for your valuable suggestion. Figure 2 and Figure 3 in the original manuscript were merged into one figure, as shown in Figure 2. The different sections of storm surge barriers along the coastline of the study area and two gauging stations including Huizhou gauging station and Gangkou gauging station were marked. The lines and dots represent the storm surge barriers and gauging stations, respectively. It has been added in the revised manuscript.

[Figure]

**Figure 2**. The two stations including Huizhou gauging station and Gangkou gauging station measure the water levels in the study area.

(6) Line 195, please provide references for these existing studies.

**Response:** Thanks for your valuable suggestion. The references (Kerr et al., 2013; Orlić et al., 2010; Li et al., 2020) about the hydrology model applied to the regions have been added to the revised manuscript.

(7) Line 204-205, could you provide further details about the 11 astronomical tidal components? And only tidal components?

**Response:** Thanks for your valuable suggestion. The open boundary of the model water level is controlled by the total water level, which is obtained by the superposition of 11 astronomical tidal components. These 11 astronomical tidal components are M2, N2, S2, K2, K1, O1, P1, Q1, MS4, M4, M6. The details have been added to the revised manuscript.

(8) Figure 4, the red rectangular in Panel a is different from the region of Panel b and c.

**Response:** Thanks for your valuable suggestion. The red rectangular spanning from 113°W to 116°W

longitude in the original Figure 4 was replaced with the rectangular spanning from 114˚W to 116˚W longitude in the revised manuscript, which is exactly corresponded to the region in the Panel b and c. The remade figure is shown in Figure 3.

[Figure]

(a)

(b)

(c)

**Figure 3.** A computational domain (a) and the domain over the study area (b) (c).

(9) In Lines 238-240: 'The Absolute Error (AE) is computed when the highest measured water level is above 100 cm. The Relative Error (RE) is calculated as the measured observed water level is below 100 cm.' Is it consistent with that in Table 2?

**Response:** Thanks for your valuable suggestion. The sentence was written wrong in Lines 238-240. The rewritten text was added in the revised manuscript that the Absolute Error (AE) is computed when the highest measured water level is below 100 cm and the Relative Error (RE) is calculated as the measured observed water level is above 100 cm, which is consistent with statistics in Table 2

(10) Please provide references to Equations in 3.2.2

**Response:** Thanks for your valuable suggestion. The references (Vickery et al., 2000; Cheung et al., 2007) on the Equations in 3.2.2 have been added in the revised manuscript.

(11) What meaning are the colors of the curves in Figure 9?

**Response:** Thanks for your valuable suggestion. Abbreviations that Tropical Depression (TD), Tropical Storm (TS), Severe Tropical Storm (STS), Typhoon (TY), Severe Typhoon Super (STY), and Super Typhoon (Super TY) are explained in detail in the Figure 9 caption. The rewritten Figure 9 caption has been added in the revised manuscript.

(12) What the meaning of the green and red dots in Figure 10?

**Response:** Thanks for your valuable suggestion. The remade Figure containing figure legend in which the different colors represent wind speeds has been added in the revised manuscript. The remade Figure is as follows.

[Figure]

**Figure 4.** The track of super typhoon Mangkhut over the western North Pacific and the South China Sea. The six categories of wind speeds are represented by the different colored dots.

(13) The 'Value' in Table 6 is a try of set up via this work, or refers to some references?

**Response:** Thanks for your valuable suggestion. The vulnerability values of land types to storm surge are provided by the guideline, which can be downloaded from the link (Ministry of Natural Resources of the People's Republic of China, 2019).

**Minor comments:**

(1) Line 6: to use 'concentrations'.

**Response:** Thanks for your correction.

(2) Line 39: should it be the word 'severe'?

**Response:** Thanks for your correction. The 'Server' has been rewritten with 'severe'.

(3) Line 44: Please use 'RMB' instead of 'Yuan' throughout the manuscript.

**Response:** Thanks for your correction. The 'Yuan' has been replaced with 'RMB'.

(4) Line 48: Please use a proper reference for IPCC report.

**Response:** Thanks for your suggestion.

(5) Line 63: impact 'on', not 'to'.

**Response:** Thanks for your correction. The 'to' has been replaced with 'on'.

(6) Line 79: Is there a reference for the 'Standard Technical Guideline'? or webpage?

**Response:** Thanks for your correction. The reference was added in the revised manuscript.

(7) Line 101: Use 'depicts' instead of 'details'.

**Response:** Thanks for your correction. The 'details' has been replaced with 'depicts'.

(8) Line 237: Use 'are' instead of 'were'.

**Response:** Thanks for your correction. The 'were' has been replaced with 'are'.

(9) In Table 5 column 1st, the bracket is not paired in format.

**Response:** Thanks for your correction.

(10) Line 527: replace 'increasing' by 'lowering'.

**Response:** Thanks for your correction. The 'increasing' has been replaced with 'lowering'

(11) Line 577: Use increased instead of heightened.

**Response:** Thanks for your correction. The 'heightened' has been replaced with 'increased'

**References**

[revised manuscript text omitted]